# Pruning for GNNs: Lower Complexity with Comparable Expressiveness

**Dun Ma** [1]  **Jianguo Chen** [2 3]  **Wenguo Yang** [3]  **Suixiang Gao** [3 4]  **Shengminjie Chen** [5]

## Abstract

In recent years, the pursuit of higher expressive power in graph neural networks (GNNs) has often led to more complex aggregation mechanisms and deeper architectures. To address these issues, we have identified redundant structures in GNNs, and by pruning them, we propose Pruned MP-GNNs, K-Path GNNs, and K-Hop GNNs based on their original architectures. We show that 1) Although some structures are pruned in Pruned MP-GNNs and Pruned K-Path GNNs, their expressive power has not been compromised. 2) K-Hop MP-GNNs and their pruned architecture exhibit equivalent expressiveness on regular and strongly regular graphs. 3) The complexity of pruned K-Path GNNs and pruned K-Hop GNNs is lower than that of MP-GNNs, yet their expressive power is higher. Experimental results validate our refinements, demonstrating competitive performance across benchmark datasets with improved efficiency.

## 1. Introduction

Currently, most GNNs (Kipf & Welling, 2017; Duvenaud et al., 2015; Hamilton et al., 2017; Veličković et al., 2018; Li et al., 2015; Zhang et al., 2018; Xu et al., 2019a) follow a framework that iteratively aggregate information from neighboring nodes and updates node representations. Since the message passing procedure is similar to the 1-dimensional Weisfeiler-Lehman (1-WL) test (Weisfeiler & Lehman, 1968), the expressive power of message-passing GNNs is also limited by the 1-WL test (Xu et al., 2019a;

Morris et al., 2019). Specifically, GNNs cannot distinguish between non-isomorphic graph structures that the 1-WL test cannot differentiate. To address this, many works extend message passing to $K$-Hop (Abu-El-Haija et al., 2019; Nikolentzos et al., 2020; Wang et al., 2021; Chien et al., 2021; Brossard et al., 2020) or $K$-Path (Zhang et al., 2023; Michel et al., 2023; Ma et al., 2020) message passing, which enhances the framework's expressive power. In $K$-Hop and $K$-Path GNNs, node representations are updated by aggregating information not only from neighbors but also from all nodes within $K$-hop and $K$-path distances. Despite the great success of GNNs in handling graph data, they still have some deficiencies: (1)Expressive power restricted by the number of layers. (2) Excessive depth hinders GNN's performance due to nonlinearity. (3) Redundant structure of GNNs. (4) Growing complexity with the increase in expressive power. Those deficiencies might impact their performance.

To address these deficiencies, we theoretically characterize the expressive power of MP-GNN, $K$-Hop, and $K$-Path message passing GNNs by utilizing matrix language tool (Brijder et al., 2019; Geerts, 2021). Surprisingly, we discovered the redundant structures in all three types of GNN frameworks. We "cut off" the redundant structures in MP-GNN, $K$-Hop, and $K$-Path message passing GNNs, and therefore propose the pruned MP-GNN, pruned $K$-Hop, and pruned $K$-Path frameworks. We prove the equivalence of expressive power between MP-GNN, $K$-Hop, and $K$-Path message-passing GNNs and the matrix language $\mathcal{L}_1 = \{., ^\top, \mathbf{1}, diag\}$, therefore we are able to demonstrate that the pruned frameworks are as powerful as the original frameworks for MP-GNN and $K$-Path message passing GNNs, and show the equivalence between pruned $K$-Hop frameworks and $K$-Hop frameworks for distinguishing regular and strongly regular graphs by utilizing the matrix language tool. Additionally, we show that the complexity of pruned $K$-Path GNNs and pruned $K$-Hop GNNs is lower than the complexity of MP-GNNs, yet they can distinguish more non-isomorphic graphs that MP-GNN cannot.

The proposed pruned frameworks have several additional advantages. (1) Unlike MP-GNNs, the range of neighbor information aggregation in the pruned MP-GNNs grows exponentially with the number of layers. Therefore, the pruned MP-GNN does not require many layers to obtain a

---

[1]School of Advanced Interdisciplinary Sciences, University of Chinese Academy of Sciences [2]Academy of Mathematics and Systems Science, Chinese Academy of Sciences [3]School of Mathematical Sciences, University of Chinese Academy of Sciences [4]Zhongguancun Laboratory.Beijing, China [5]State Key Lab of Processors, Institute of Computing Technology, Chinese Academy of Sciences. Correspondence to: Wenguo Yang <yangwg@ucas.ac.cn>.

*Proceedings of the $42^{nd}$ International Conference on Machine Learning*, Vancouver, Canada. PMLR 267, 2025. Copyright 2025 by the author(s).

vast amount of neighbor information. (2) Pruned MP-GNNs do not need to pile up many layers as MP-GNNs, hence the nonlinearity which hinders GNN's performance of node representations is restricted. (3) In our pruned framework, the redundant structure of GNNs has been "cut off", therefore the framework is easier to train and optimize. (4) In large graphs, our pruned $K$-Path and $K$-Hop GNNs, compared to MP-GNNs, can distinguish more non-isomorphic graphs that MP-GNNs cannot while maintaining the lower complexity. We conduct both synthetic and real experiments, which demonstrate the superior performance of our proposed pruned framework and validate our theoretical findings.

## 2. Preliminaries

In this section, we present some basic notations and concepts of the matrix language and the message passing framework.

### 2.1. Notations

Denote a graph as $G = (V, E)$, where $V = \{1, 2, ..., n\}$ is the node set and the edge set is denoted as $E \subseteq V \times V$. The adjacency matrix is denoted by $A \in \{0, 1\}^{n \times n}$. The notation $[m, n]$ represents the integers from $m$ to $n$, and $[n]$ represents $[1, n]$.

We define the $k$-length walk $(v, u_1, \cdots, u_k)$ as $k$-walk neighbor of $v$, its representation is the same as that of node $u'_k s$. Similarly, we can define $k$-path neighbor of $v$, if $(v, u_1, \cdots, u_k)$ is a path. The set of $k$-walk\path neighbors is denoted as $N^k_{walk}(v) \backslash N^k_{path}(v)$. Note that the element $A^k(i, j)$ equals the number walk from $v_i$ to $v_j$ with length k. Hence, we denote $A^k$ as graph $G's$ $k$-walk adjacency matrix. Denote $P_k \in N^{n \times n}$ as $k$-path neighbor matrix. The element $P_k(i, j)$ represents the number of k-length paths from $v_i$ to $v_j$. Node $u$ is called node $v's$ $k$-hop neighbor: $u \in N^k_{hop}(v)$, if the length of shortest path from $v$ to $u$ equal to $k$. Denote $O_k$ as the k-hop matrix, it's worth noticed that $O_k \in \{0, 1\}^{n \times n}$

$\mathcal{A}$ is a graph isomorphism algorithm, $(G, G')$ is a pair of graphs and $L \in \mathbb{N}^+$, we denote $(G, G') \in GI^L_{\mathcal{A}}$, if algorithm $\mathcal{A}$ decides $(G, G')$ is isomorphic at $L^{th}$ iteration. In this paper, we assume that the aggregation and combination functions of all the mentioned GNNs are injective, making their expressive power equivalent to the corresponding WL Test. In this paper, we only consider necessary graph isomorphism algorithms, which means that the isomorphism to algorithm is only a necessary but not sufficient condition for graph isomorphism. Therefore, given two graph isomorphism algorithm, $\mathcal{A}$ and $\mathcal{B}$, if $GI^L_{\mathcal{A}} \subseteq GI^L_{\mathcal{B}}$, then algorithm $\mathcal{A}$ is regarded to be more powerful than $\mathcal{B}$.

### 2.2. Matrix Language

Recently, Brijder (Brijder et al., 2019) and Geerts (Geerts, 2021) proposed a new matrix language called MATLANG. Matrix languages can be formalised through composition of linear algebra operations. Intuitively, a linear algebra operation takes a number of matrices as input and returns another matrix(or vector or scalar). We employ MATLANG as a theoretical tool for our proofs, with its detailed formulation provided in Appendix D.

### 2.3. Graph Neural Networks

The WL algorithm is a heuristic method for graph isomorphism testing that iteratively refines vertex labels based on their neighbors' labels, which is specifically introduced in Appendix B. MP-GNNs match the expressive power of WL when their feature aggregation and combination functions are injective. Most GNNs use a message passing framework, where the features of a node's neighbors are first aggregated and then combined with the node's own features. In this subsection, denote $H^l_v$ as the output representation of node $v$ at layer $l$.

**Message Passing Framework.** In the standard message passing framework, the update rules are typically written as:

$$M^l_v = AGG^l(\{\!\{H^{l-1}_u | u \in N(v)\}\!\}) \tag{1}$$

$$H^l_v = COB^l(H^{l-1}_v, M^l_v) \tag{2}$$

where $M^l_v$ represents the message received by node $v$ at layer $l$, $AGG^l$ and $COB^l$ are the aggregation and combination functions, respectively. After $L$ layers of message passing, $H^L_v$ is the final representation of node $v$. According to Theorem 3 in (Xu et al., 2019a), nodes receive the same feature representations if and only if they receive the same labels in the corresponding WL test.

$K$**-Path Message Passing Framework.** The $K$-Path GNN extends the standard message passing to consider paths of length $k$ ($1 \leq k \leq K$). Specifically, each $k$-path neighbor set $N^k_{path}(v)$ is aggregated independently and then combined. Specifically,

$$M^{l,k}_v = AGG^l_k(\{\!\{H^{l-1}_u \mid u \in N^k_{path}(v)\}\!\}), \tag{3}$$

$$\mathbf{M}^l_v = (M^{l,1}_v, M^{l,2}_v, \cdots, M^{l,K}_v) \tag{4}$$

$$H^l_v = COB^l(H^{l-1}_v, \mathbf{M}^l_v) \tag{5}$$

A node can appear multiple times as a $k$-path neighbor of $v$, allowing the algorithm to capture repeated walks of length $k$. We use $(G, G') \in GI^L_{K-P}$ to denote that the $K$-Path WL test decides $G$ and $G'$ are isomorphic after $L$ iterations.

$K$**-Hop Message Passing Framework.** The $K$-Hop GNN differs by focusing on shortest-path neighbors. Each node

$u$ belongs to $N_{hop}^k(v)$ if the shortest path from $v$ to $u$ has length $k$. Importantly, a node can be a $k$-hop neighbor of $v$ at most once. The update process is similar to the $K$-Path Message Passing framework, but the aggregation process equation 3 is replaced by the following steps:

$$M_v^{l,k} = AGG_k^l\big(\{\!\{H_u^{l-1} \mid u \in N_{hop}^k(v)\}\!\}\big) \qquad (6)$$

Likewise, we denote by $(G, G') \in GI_{K-H}^L$ the event that the $K$-Hop WL test deems $G$ and $G'$ isomorphic after $L$ iterations.

By introducing $K$-path or $K$-hop neighbors, these frameworks can potentially enhance the expressive power of GNNs relative to the standard 1-Hop message passing, albeit with higher computational cost.

## 2.4. Deficiencies of Graph Neural Networks

Despite the great success of GNNs in handling graph data, they still have some deficiencies that might impact their performance as follows.

**Expressive power restricted by the number of layers**: An MP-GNN with $L$ layers can aggregate information from at most $L$-walk neighbors. Therefore, the expressive power of GNNs is particularly limited by the number of layers, especially on large graphs.

**Excessive depth hinders GNN's performance.** Suzuki (2020) points out piling up many layers does not improve GNN's performance (or sometimes worsens it). The core reason is that each layer's iteration increases the nonlinearity of node representations, causing the differences between node representations to diminish over layers.

**Redundant structure of GNN**: The redundant structure of GNNs will lead to reduced computational efficiency, an increased search space during optimization, and, in some cases, a decline in the model's generalization ability. In the section 4, we will demonstrate the redundant structure in GNNs.

**Growing Complexity with the increase in expressive power**: Since $Xu\,et\,al.$ (2019b) pointed out that MP-GNNs might suffer from insufficient expressive power, many GNN variants with higher expresive power have been proposed. However, as their expressive power increases, both time and space complexity usually also grows.

## 3. Pruned Message Passing Framework

In order to address the shortcomings of GNNs mentioned in previous section, we propose pruned frameworks for MP-GNN and its variants, including $K$-path and $K$-hop message passing frameworks. These pruned frameworks have streamlined structures, resulting in lower computational complexity compared to the original frameworks.

## 3.1. Pruned ($a_k$-walk) Message Passing Framework

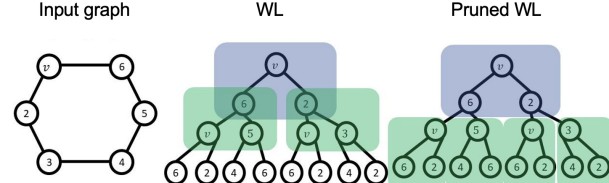

*Figure 1.* Comparison of MP-GNN and pruned MP-GNN: Blue and Green represents structure information node $v$ has aggregated at $1^{st}$ and $2^{nd}$ iterations respectively. The overlap in the MP-GNN indicates computational redundancy.

Given a sequence of positive integers $a_k$, we propose a new framework called the $a_k$-walk message passing framework like as Fig. 1. The key difference from the standard MP framework is that, at the $k^{th}$ layer, node $v$ aggregates features from its $a_k$-walk neighbors, rather than just its direct neighbors. Specifically, $H_v^k$ is computed as follows:

$$M_v^k = AGG^k(\{\!\{H_u^{k-1} \mid u \in N_{walk}^{a_k}(v)\}\!\}) \qquad (7)$$

$$H_v^k = COB^k(H_v^{k-1}, M_v^k) \qquad (8)$$

We point out the aggregation of $k$-walk neighbors' features can be computed as

$$M_v^k = AGG^k(\{\!\{H_u \mid u \in N_{walk}^{a_k}(v)\}\!\}) = \underbrace{AGG(\cdots AGG}_{a_k\ times}(\{\!\{H_u \mid u \in N(v)\}\!\})). \qquad (9)$$

Hence the computation complexity of $AGG(\{\!\{H_u \mid u \in N_{walk}^{a_k}(v)\}\!\})$ will not exceed $a_k$ times of the computational complexity of $AGG(\{\!\{H_u \mid u \in N(v)\}\!\})$. We provide a long-refinement graph example for comparison in Appendix $C.4$.

We refer to the $a_k$-walk message passing framework when $a_k = 2^{k-1}$ as the pruned message passing framework. Additionally, we denote $(G, G') \in GI_{Pr}^L$ if the pair of graphs $(G, G')$ gets the same representation in pruned MP-GNN frameworks. We also provide Pruned Message Passing Framework's corresponding WL Test in $C.1$.

## 3.2. Pruned $K$-Path Message Passing Framework

In Figure 2, the $K$-path message passing framework exhibits redundant computations as certain structural information is repeatedly encoded into nodes' features across iterations. To address this, we propose the pruned $K$-path message passing framework. Unlike the standard $K$-path framework, the pruned version aggregates features from the $l$-path neighbors $N_{path}^l$ to the $K$-path neighbors $N_{path}^K$ at the $l^{th}$ layer ($l \leq K$), rather than from the 1-path neighbors $N_{path}^1$ to the

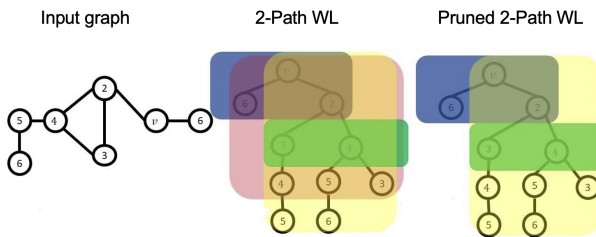

Figure 2. Comparison of 2-path and pruned 2-path frameworks, where colors denote aggregated structure information for node $v$: Blue ($1^{st}$ iteration, 1-path neighbors), Green ($1^{st}$ iteration, 2-path neighbors), Red ($2^{nd}$ iteration, 1-path neighbors), and Yellow ($2^{nd}$ iteration, 2-path neighbors). The range of Red is covered by Blue, Green, and Yellow.

$K$-path neighbors $N_{path}^K$. Let $H_v^l$ denote the feature of node $v$ at the $l^{th}$ layer. When $l \leq K$, the pruned $K$-path message passing framework is defined as follows:

$$M_v^{l,k} = AGG_k^l(\{\{H_u^{l-1}|u \in N_{path}^k(v)\}\})(k \leq l \leq K) \quad (10)$$

$$\mathbf{M}_v^l = (M_v^{l,l}, M_v^{l,l+1}, \cdots, M_v^{l,K}) \quad (11)$$

$$H_v^l = COB^l(H_v^{l-1}, \mathbf{M}_v^l) \quad (12)$$

else when $l > K$, the $K$-path pruned Message passing framework can be defined as follows:

$$M_v^{l,K} = AGG_K^l(\{\{H_u^{l-1}|u \in N_{path}^K(v)\}\})$$
$$H_v^l = COB^l(H_v^{l-1}, M_v^{l,K}) \quad (13)$$

Denote $(G, G') \in GI_{PR\ K-P}^L$, if the pruned $K$-Path framework's corresponding WL test decides $G$ and $G'$ are isomorphic after L iterations.

### 3.3. Pruned $K$-Hop Message Passing Framework

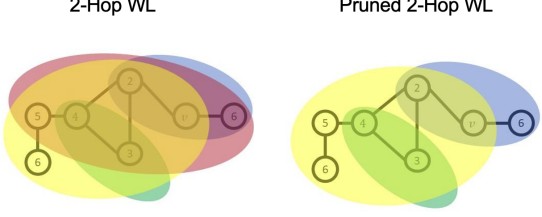

Figure 3. Comparison of 2-hop and pruned 2-hop frameworks. Colors indicate aggregated structure information for node $v$: Blue ($1^{st}$ iteration, 1-hop neighbors), Green ($1^{st}$ iteration, 2-hop neighbors), Red ($2^{nd}$ iteration, 1-hop neighbors), and Yellow ($2^{nd}$ iteration, 2-hop neighbors). The range of Red is covered by Blue, Green, and Yellow.

In figure 3, at the iteration of $K$-hop message passing framework, there's also specific node structure information which

has been repetitively encoded into nodes' features which will cause redundant computations. Therefore, we propose the pruned $K$-hop message passing framework. Different from $K$-hop message passing framework, when $l \leq K$, the $K$-hop pruned message passing framework is similar to the $K$-path pruned message passing framework, but the aggregation process equation 10 is replaced by the following:

$$M_v^{l,k} = AGG_k^l(\{\{H_u^{l-1}|u \in N_{hop}^k(v)\}\}) \quad (14)$$

else when $l > K$, the $K$-hop pruned message passing framework can be defined as follows:

$$M_v^{l,K} = AGG_K^l(\{\{H_u^{l-1}|u \in N_{hop}^K(v)\}\})$$
$$H_v^l = COB^l(H_v^{l-1}, M_v^{l,K}). \quad (15)$$

Denote $(G, G') \in GI_{RE\ K-H}^L$, if the pruned $K$-Hop framework's corresponding WL test decides $G$ and $G'$ are isomorphic after $L$ iterations. We also provide $k$-path GNN's corresponding WL Test in *Appendix C.3*.

## 4. Pruned Framework's Equivalent Expressiveness Solving GNNs' Deficiencies

In this section, we theoretically analyze the expressive power of the message passing framework, $K$-Path and $K$-Hop frameworks and their pruned frameworks. We also demonstrate how our pruned frameworks

We assume there are no edge features and all nodes in the graph have the same feature, which means that GNNs can only distinguish two nodes based on local structure of nodes. Let aggregation and combination functions be injective. We will show that the pruned message passing framework is as powerful as 1-WL test and pruned $K$-Path is as powerful as $K$-Path framework. As for pruned $K$-Hop framework, it is as powerful as $K$-Hop framework when distinguishing regular or strong regular graphs.

### 4.1. Expressive power of Pruned Message Passing Framework

In this subsection, we delve into a theoretical analysis of the expressive power of $a_n$-walk (pruned) message passing framework. Xu et al. (2019a) have proven that the expressive power of 1-hop message passing is bounded by the 1-WL test.

Given a positive integers sequence $a_k$, $S_k = \sum_{t \in [k]} a_t$. We say that a sequence $a_k$ is *viewable* if $\forall k \in \mathbb{N}^+, r \in [S_k]$, there exists a subset $T \subseteq [k]$, such that $\sum_{t \in T} a_t = r$. We will prove that if $a_k$ is viewable, then $a_k$-walk message passing framework is at least as powerful as 1-WL test. The theorem is as follows:

**Theorem 4.1.** *Given a positive integer sequence $a_k$ and a pair of graphs $(G, G')$, $S_k = \sum_{t \in [k]} a_t$, if $a_k$ is viewable,*

*then* $\forall l \in \mathbb{N}^+ \ GI^l_{a_k-walk} \subseteq GI^{S_l}_{WL}$.

The following lemma and theorem are meant to explain why we chose $2^{k-1}$-walk as the pruned framework:

**Lemma 4.2.** *Given a positive integer sequence* $a_k$, $S_k = \sum_{t \in [k]} a_t$, *if* $S_k > 2^k - 1$, *then* $a_k$ *is not viewable.*

**Theorem 4.3.** *Given a positive integer sequence* $a_k$, *if* $a_k$ *is not viewable, then there's a pair of non-isomorphic graphs* $(G, G')$ *that* $(G, G') \in GI^l_{a_k-walk\,WL}$ *but* $(G, G') \notin GI^{S_l}_{WL}$.

The proof of *Lemma* 4.2, *Theorem* 4.1 and *Theorem* 4.3 is provided in *Appendix E*. *Lemma* 4.2 indicates that when $a_k = 2^{k-1}$, the pruned MP-GNN framework can collect information from the most distant nodes while maintaining an expressive power comparable to that of MP GNNs. *Theorem* 4.3 provides an example to demonstrate that if $a_k$ is not viewable, then the expressive power of $a_k$-walk framework will be diminished.

Despite the expressive power of $a_n$-walk message passing framework is bounded by 1-WL test, we are able to "cut off" the number of GNN's layers and parameters to reduce computational complexity.

We now show why $2^{k-1}$-walk message passing framework is more effective than message passing framework. As the proof of *Theorem* 4.1 shows, for an unlabeled graph, the determining factor of label that node $v$ gets in 1-WL test is the cardinality of $v's$ $l$-walk neighbor $|N^l_{walk}(v)|$. As *Figure* 1 suggests, there's an overlapping part between the blue rectangle and green rectangles, which implies cardinality information has been repetitively encoded into node $v$. However, the pruned WL test, as the figure shows, makes sure that every $l$-walk neighbor's cardinality information will be encoded into node $v$ without repetition. This makes the pruned framework more effective than message passing framework.

### 4.2. Expressive Power of Pruned $K$-Path and $K$-Hop Message Passing Framework

In this subsection, we conduct a detailed analysis of the expressive power of $K$-Path and $K$-Hop GNNs and their pruned frameworks. We will show that, although the pruned $K$-Path GNNs message passing framework does not aggregate as much graph structure information as $K$-Path framework, this does not diminish the expressive power of pruned $K$-Path framework. The theorem is as follows:

**Theorem 4.4.** *Given a pair of graphs* $(G, G')$ *and* $K \in \mathbb{N}^+$, *the expressiveness of pruned $K$-Path framework is as powerful as $K$-Path framework. In other words,* $\forall L \in \mathbb{N}^+$ $GI^L_{PR\,K-P} = GI^L_{K-P}$

The proof of *Theorem* 4.4 is provided in the Appendix.

We indicate the core of the proof of *Theorem* 4.4 is transforming the expressiveness problem of $K$-Path GNN into an algebraic problem through MATLANG. Specifically, we identify a matrix language that is expressively equivalent to the $K$-Path framework and proved that the pruned $K$-Path framework has the same expressiveness as this matrix language.

As for the pruned $K$-Hop framework, unfortunately, we have to point out the difficulty in proving the equivalent expressive power between the pruned $K$-Hop framework and $K$-Hop framework. We emphasize the main reason can be traced back to the proof of *Theorem E*.10 in the Appendix: during the process of generating $K$-hop matrices from $K$-path matrices, the matrix language has removed two critical structural pieces of information (1): the number of shortest paths between two nodes. (2)the number of paths which are longer than shortest paths between two nodes.

On the other hand, the $K$-Hop GNN was proposed largely to compensate for the inability of MP-GNN to distinguish regular graphs. Let $RG$ denote the set of pairs of regular graphs, and $SRG$ denote the set of pairs of strong regular graphs. We show that pruned $K$-Hop GNN has comparable expressiveness as $K$-Hop GNN in distinguishing strong regular graphs, as well as for for regular graphs when $K = 2$. In other words:

**Theorem 4.5.** $\forall L, K \in N^+, (RG \cap GI^L_{PR\,2-H}) \subseteq (RG \cap GI^L_{2-H})$ *and* $(SRG \cap GI^L_{RE\,K-H}) \subseteq (SRG \cap GI^L_{K-H})$.

### 4.3. Pruned Frameworks Address Deficiencies of GNNs

In this subsection, we demonstrate how pruned frameworks address deficiencies of GNNs. Regarding the limited expressive power restricted by the number of layers, unlike MP-GNNs the range of neighbor information aggregation in the pruned MP-GNNs grows exponentially with the number of layers. Therefore, the pruned MP-GNN does not require many layers to obtain a vast amount of neighbor information. Meanwhile, the pruned MP-GNNs do not need to pile up as many layers as MP-GNNs, thus the nonlinearity of node representations is restricted.

We now show the redundant structure of MP-GNNs along with a comparison to pruned MP-GNNs through a example. Given a connected graph with at least 3 nodes, while some nodes are specially marked, we consider a simple node-level task for GNNs. We want node $v$'s 1-dimensional representation $H_v$ outputs 1 if and only if $v$ is marked or there's node $u$ marked while the distance between $v$ and $u$ is less than or equal to 3, and outputs 0 otherwise. We consider a 3-layer MP-GNN $\mathcal{M}$, the initial representation $H^0_v = 1$ if and only if $v$ is marked and $H^0_v = 0$ otherwise, its representation is

*Table 1.* Simulation dataset result. The best is highlighted.

| Method | Node Properties | | | Graph Properties | | | Counting Substructures (MAE) | | | |
|---|---|---|---|---|---|---|---|---|---|---|
| | SSSP | Ecc. | Lap. | Connect. | Diameter | Radius | Tri. | Tailed Tri. | Star | 4-Cycle |
| **GIN** | -2.1476 | -1.9038 | -1.6000 | -1.9239 | -3.3079 | -4.7584 | 0.3306 | 0.1534 | -0.8716 | 0.1176 |
| **PR GIN** | -1.9202 | -1.6817 | -1.7772 | -1.7342 | -3.2023 | -4.9635 | 0.2893 | 0.3737 | -0.9309 | 0.1260 |
| $K$-**Path** | -2.7063 | -2.4900 | -4.9596 | -4.3159 | -3.8475 | -5.2038 | -1.3566 | -1.2709 | -0.9342 | -0.4936 |
| **PR** $K$-**Path** | -2.7833 | -2.5977 | **-5.284** | **-4.4239** | -3.9324 | **-5.2983** | **-1.4588** | **-1.3839** | -1.0391 | **-0.6070** |
| $K$-**Hop** | **-2.8152** | -2.5963 | -4.6254 | -2.1877 | **-3.9683** | -5.2923 | 0.1919 | 0.0637 | -1.0043 | 0.0960 |
| **PR** $K$-**Hop** | -2.7785 | **-2.6672** | -4.9970 | -2.1472 | -3.7005 | -5.0619 | 0.0499 | 0.0091 | **-1.3141** | 0.0665 |

*Table 2.* TU dataset (Yanardag & Vishwanathan, 2015) evaluation result. The top three are highlighted with bold text and marked in red, blue, and black, respectively.

| Method | MUTAG | D&D | PTC-MR | PROTEINS | IMDB-B |
|---|---|---|---|---|---|
| **WL** | 90.4±5.7 | 79.4±0.3 | 59.9±4.3 | 75.0±3.1 | 73.8±3.9 |
| **GraphSAGE** | 91.7±6.5 | 78.1±2.6 | 66.5±4.0 | 76.5±4.6 | 76.4±2.7 |
| **GraphSNN** | 91.2±2.5 | 82.4±2.7 | 66.9±3.5 | 76.5±2.5 | **76.9±3.3** |
| **GIN** | 88.4±5.2 | 76.3 ±1.2 | 63.6 ±5.0 | 74.9 ±1.9 | 74.1±3.5 |
| **PR GIN** | 89.7±3.0 | 77.2 ±0.3 | 64.7 ±1.3 | 75.8 ±2.0 | 75.2±3.4 |
| $K$-**Hop** | 92.1±2.0 | 81.9±3.4 | 64.2±2.5 | 75.0±1.2 | 74.5± 1.0 |
| **PR** $K$-**Hop** | 91.8±4.3 | **84.1 ±0.9** | 65.5 ±4.1 | 77.0±3.4 | 76.2±0.6 |
| $K$-**Path** | 91.7±1.7 | 82.1±3.4 | 65.9±5.1 | 78.6±1.8 | 76.8±3.2 |
| **PR** $K$-**Path** | **92.6 ±3.3** | 83.3 ±0.4 | **67.5 ±3.5** | **79.1 ±2.1** | 74.9±2.9 |

updated as follows:

$$H_v^l = \sigma(H_v^{l-1} \cdot W_1^l + \sum_{u \in N(v)} H_u^{l-1} \cdot W_2^l), \quad (16)$$

where $\sigma(x) = \min(\max(0, x), 1)$ parameter matrix $W_i^l \in \{(0), (1)\}(i \in [2], l \in [3])$. There are 4 parameter configurations for $\mathcal{M}$, one of them is setting all $W_i^l = 1$, while the remaining three are to set one of these parameter configurations $W_1^l = 0$ for $l \in [3]$. We prove that among all the parameter settings of $\mathcal{M}$, only these four are capable of handling this task in the Appendix. In other words, we can randomly choose one of the parameters $W_1^l$ equal to 0, the structure of MP-GNN will represent the same outcome. We can observe the redundant structure of GNNs from the randomness in the selection of $W_1^l$.

However, for the pruned MP-GNN, we consider 2-layer $\mathcal{M}'$ as

$$H_v^l = \sigma(H_v^{l-1} \cdot W_1^l + \sum_{u \in N_{walk}^{2l-1}(v)} H_u^{l-1} \cdot W_2^l). \quad (17)$$

Except for setting all parameters equal to 1, there is only one parameter configuration for $\mathcal{M}'$, as setting $W_1^2 = 0$, which means that we have pruned this redundant structure, reducing the cardinality of the parameter space from $2^6$ in MP-GNN to $2^4$ in pruned MP-GNN.

As for the high expressive power accompanied with higher complexity, we refer our pruned framework: pruned $K$-Path and $K$-Hop GNN. When layer $l > K$, the complexity of pruned $K$-Path and $K$-Hop GNN is the same as MP-GNN, but the breadth of neighbor information obtained in each iteration of pruned $K$-Path and $K$-Hop GNN is equivalent to that of MP-GNN after $K$ iterations. Hence in a large graph, $K$-Path and $K$-Hop GNN can reduce complexity by decreasing the number of layers while still distinguishing graphs that MP-GNN cannot.

To demonstrate the efficiency of the pruned framework, we provide the iteration process of the WL Test (MP-GNN), the pruned WL Test (PR MP-GNN), and the pruned 2-Hop WL Test (PR 2-Hop GNN) on the long-refinement graph in *Appendix C.4*. We can see both pruned WL Test and pruned 2-Hop WL Test terminate at $4^{th}$ iteration while WL Test at the $11^{th}$ iteration.

## 5. Time and Space Complexity

In this section, we discuss the time and space complexity of MP, $K$-path, $K$-hop, and their pruned frameworks. We assume a graph has $n$ nodes, every framework is designed to gather nodes information at a distance of $L$, and we assume $(n \gg L \gg K)$. The time and space complexities are listed

*Table 3.* QM9 results. The best is highlighted.

| Target | GraphSAGE | GIN | PR GIN | $K$-Hop | PR $K$-Hop | $K$-Path | PR $K$-Path |
|---|---|---|---|---|---|---|---|
| $\mu$ | 0.369 | 0.355 | 0.422 | 0.301 | 0.303 | 0.311 | **0.268** |
| $\alpha$ | 0.308 | **0.258** | 0.277 | 0.384 | 0.282 | 0.274 | 0.291 |
| $\varepsilon_{\text{HOMO}}$ | 0.00382 | 0.00427 | 0.00335 | 0.00266 | 0.00276 | 0.00221 | **0.00197** |
| $\varepsilon_{\text{LUMO}}$ | 0.00492 | 0.00644 | 0.00264 | 0.00294 | **0.00275** | 0.00281 | 0.00279 |
| $\Delta\varepsilon$ | 0.01187 | 0.00419 | 0.00430 | 0.00379 | 0.00396 | 0.00357 | **0.00316** |
| $\langle R^2 \rangle$ | 16.38 | 20.97 | 20.69 | 16.13 | 20.44 | 17.66 | **15.70** |
| ZPVE | 0.001747 | 0.001262 | 0.001537 | 0.000220 | **0.000141** | 0.000174 | 0.000159 |
| $U_0$ | 2.05 | 2.05 | 2.31 | 0.0690 | 0.0629 | 0.0571 | **0.0502** |
| $U$ | 2.05 | 2.02 | 2.00 | 0.0650 | 0.0593 | 0.0637 | **0.0527** |
| $H$ | 2.05 | 2.02 | 2.00 | 0.0589 | 0.0604 | 0.0533 | **0.0552** |
| $G$ | 2.05 | 2.02 | 2.00 | 0.0663 | 0.0574 | **0.0543** | 0.0559 |
| $C_v$ | 0.2716 | 0.2170 | 0.2262 | **0.0841** | 0.0847 | 0.0878 | 0.0847 |

*Table 4.* Verification on the equivalence of the expressive power.

| Method | K | EXP (ACC) | | SR (ACC) | | CSL (ACC) | |
|---|---|---|---|---|---|---|---|
| | | SPD | GD | SPD | GD | SPD | GD |
| **GIN** | | 50 | 50 | 6.67 | 6.67 | 12 | 12 |
| **PR GIN** | | 50 | 50 | 6.67 | 6.67 | 12 | 12 |
| | K=2 | 50 | 50 | 73.33 | 73.33 | 52.7 | 52.7 |
| $K$-**Path** | K=3 | 100 | 100 | 73.33 | 73.33 | 90 | 90 |
| | K=4 | 100 | 100 | 73.33 | 73.33 | 100 | 100 |
| | K=2 | 100 | 100 | 73.33 | 73.33 | 52.7 | 52.7 |
| **PR $K$-Path** | K=3 | 100 | 100 | 73.33 | 73.33 | 90 | 90 |
| | K=4 | 100 | 100 | 73.33 | 73.33 | 100 | 100 |
| | K=2 | 50 | 50 | 6.67 | 6.67 | 32 | 22.7 |
| $K$-**Hop** | K=3 | 100 | 66.9 | 6.67 | 6.67 | 62 | 42 |
| | K=4 | 100 | 100 | 6.67 | 6.67 | 92.7 | 62.7 |
| | K=2 | 50 | 50 | 13.33 | 13.33 | 32 | 22.7 |
| **PR $K$-Hop** | K=3 | 100 | 66.9 | 13.33 | 13.33 | 62 | 62.7 |
| | K=4 | 100 | 100 | 13.33 | 13.33 | 62 | 62.7 |

*Table 5.* Time and Space Complexity

| Method | Time | Space |
|---|---|---|
| **MP** | $\Theta(n \cdot L)$ | $\Theta(n \cdot L)$ |
| **PR MP** | $\Theta(n \cdot \log(L))$ | $\Theta(n \cdot \log(L))$ |
| $K$-**Hop** | $\Theta(n \cdot L)$ | $\Theta(n \cdot L)$ |
| **PR $K$-Hop** | $\Theta(n \cdot \frac{L}{K})$ | $\Theta(n \cdot \frac{L}{K})$ |
| $K$-**Path** | $\Theta(n \cdot L)$ | $\Theta(n \cdot L)$ |
| **PR $K$-Path** | $\Theta(n \cdot \frac{L}{K})$ | $\Theta(n \cdot \frac{L}{K})$ |

2021), and (3) CSL, comparing them with their original frameworks (Murphy et al., 2019). We use node properties (such as single-source shortest path, eccentricity, and Laplacian feature) and graph property regression (connectivity, diameter, radius), as well as graph substructure counting (triangle, tailed triangle, star, and 4-cycle) to demonstrate expressive power.

we also perform the original and pruned WL test on each graph to test whether pruned WL test produces the same number of node classes as the original WL test we conclude that the pruned WL test is consistent with the original WL test on that graph. The WL Test algorithm applies the hashed and power iterated color refinement algorithm proposed by Kersting K(Kersting et al., 2014), and others are derived from it. The algorithm is provided in appendix $G$:

To verify whether pruning improves the frameworks' performance, we evaluate the pruned frameworks' performance on 8 real-world datasets: MUTAG (Debnath et al., 1991), DD (Dobson & Doig, 2003), PROTEINS (Dobson & Doig, 2003), PTC-MR (Toivonen et al., 2003), and IMDB-B (Yanardag & Vishwanathan, 2015) from TU database, as well as QM9 (Ramakrishnan et al., 2014; Wu et al., 2018) and ZINC (Dwivedi et al., 2020) for molecular property prediction. The results are shown

in $Table\ 5$

Further discussion is provided in $Appendix\ F$.

## 6. Experiment

In this section, we conduct extensive experiments to evaluate the performance of the pruned frameworks. Specifically, we focus on answering the following three questions:

- **Q1**: Do the pruned frameworks have the same expressive power as the original frameworks?

- **Q2**: Does the pruning improve the frameworks' performance?

To verify the expressive power of the pruned frameworks, we empirically evaluate them on three simulation datasets: (1) EXP (Abboud et al., 2021), (2) SR25 (Balcilar et al.,

in Tables 1 to 3 and 6. And all the experimental materials in provided in `https://anonymous.4open.science/r/PrunedGNN-AC61/README.md`

To assess the improvement in the effectiveness of the pruned frameworks, we compare the running time and the number of parameters of the pruned frameworks with the original ones. Detailed dataset statistics and results are provided in the Appendix $G$.

**Verification of Equivalence of Expressive Power:** To verify the equivalence of expressive power, we evaluate both pruned and original frameworks, implementing GIN as the base encoder for each. The results in Table 4 lead to the following conclusions: (1) For both $K$-Path and $K$-hop frameworks, expressive power improves as $K$ increases. (2) The expressive power of pruned frameworks is comparable to that of the original frameworks, as evidenced by their equally excellent performance. (3) While ensuring equivalent expressive power, the pruning framework significantly improves efficiency compared to the original.

**Effectiveness on Node/Graph Properties and Substructure Prediction:** We compare the pruned frameworks with their original counterparts to evaluate their effectiveness and expressive power on node/graph properties and substructure prediction. GIN is used as the base encoder for all frameworks. Pruned $K$-hop and $K$-path frameworks achieve the best performance on most tasks. Moreover, the performance between pruned and original frameworks is highly consistent, indicating their equivalent expressive power.

*Table 6.* ZINC result.

| Method | # param. | test MAE |
|---|---|---|
| **GraphSAGE** | 480805 | 0.143±0.01 |
| **GIN** | 356406 | 0.134±0.007 |
| **PR GIN** | 256406 | 0.136±0.012 |
| $K$-**Hop** | 574613 | 0.079±0.015 |
| **PR** $K$-**Hop** | 476615 | **0.077±0.009** |
| $K$-**Path** | 581659 | 0.082±0.011 |
| **PR** $K$-**Path** | 456414 | 0.079±0.004 |

**Evaluation on TU and QM9 Datasets:** For the TU datasets, we select the graph kernel-based method (WL subtree kernel) and advanced GNN methods (GraphSNN (Wijesinghe & Wang, 2022) and GraphSAGE (Hamilton et al., 2017)) as baseline models. For both the pruned and original frameworks, we use GIN (Xu et al., 2019a) as the base encoder. The results, shown in Table 3, indicate that the pruned frameworks achieve the best performance on almost all tasks. For the QM9 dataset, except for $C_v$, both $K$-path and $K$-hop pruned frameworks outperform the others across all tasks. The ZINC dataset results, shown in Table 5, reveal that $K$-

hop pruned frameworks achieve the best performance, and the pruned frameworks perform as well as the originals but with fewer parameters. Overall, on all real-world datasets, pruned frameworks outperform their original counterparts.

## 7. Conclusion

In this work, we proposed pruned versions of MP-GNN, $K$-Hop, and $K$-Path frameworks by removing redundant structures, thereby improving efficiency. Our theoretical analysis confirmed that these pruned MP-GNN and $K$-Hop frameworks retain the expressive power of their original counterparts, with the added benefit of reduced computational complexity, and K-hop MP-GNNs and their pruned architecture exhibit equivalent expressiveness on regular and strongly regular graph. Specifically, the pruned frameworks use fewer parameters, making them more computationally efficient and easier to optimize and train. Through extensive experiments, we demonstrated that the pruned frameworks outperform the original ones in terms of runtime and memory usage, while maintaining competitive performance on various benchmark datasets. These results validate the practical advantages of using pruned frameworks without sacrificing expressive power.

## Impact Statement

This paper presents work whose goal is to advance the field of Machine Learning. There are many potential societal consequences of our work, none which we feel must be specifically highlighted here.

## Acknowledgements

This work is supported by the National Key R&D Program of China under grant 2022YFA1003900 and National Natural Science Foundation of China under Grant T2341006.

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

# Appendix

## A. Related Work

Following the pioneering studies of Xu et al. (2019b); Morris et al. (2019), a substantial body of research has focused on designing novel GNN architectures that surpass the expressiveness of the 1-WL test. Broadly, these approaches can be categorized as follows.

**Higher-order GNNs**. Unlike traditional GNNs, higher-order GNNs compute representations for multi-tuples of nodes and employ tensor operations to aggregate features across these tuples (Morris et al., 2019; Maron et al., 2019b;c;a; Keriven & Peyré, 2019; Azizian & Lelarge, 2021; Geerts & Reutter, 2022). However, the computational complexity of higher-order GNNs increases exponentially with the number of tuples, making them unsuitable for large-scale graphs. This challenge has motivated research into localized variants of higher-order GNNs, which leverage the sparse and local nature of graphs to reduce complexity at the cost of some expressiveness. Notable models include the 1-2-3 GNN (Morris et al., 2019), PPGN (Maron et al., 2019a), and Ring-GNN (Chen et al., 2019), which aim to approximate higher-order WL tests while maintaining computational efficiency. For example, Morris et al. (2020; 2022) introduced methods that localize $k$-WL aggregation to exploit graph sparsity (see Appendix B.2), while Vignac et al. (2020) proposed a localized 2-order GNN whose expressive power aligns with 3-IGN (Maron et al., 2019c). These developments illustrate how localized higher-order GNNs strike a balance between expressiveness and scalability, enabling their practical use in real-world applications.

**Substructure-based GNNs**. Chen et al. (2020) showed that standard message passing GNNs fail to detect or count common substructures such as cycles, cliques, and paths. Building on this insight, another approach to enhancing GNN expressiveness involves leveraging substructure information within graphs. For example, Bouritsas et al. (2022) proposed the Graph Substructure Network (GSN), which integrates substructure counting into node features via a preprocessing. This idea was further extended by Barceló et al. (2021) through homomorphism counting. Additional progress has been made by Bodnar et al. (2021b;a); Thiede et al. (2021); Horn et al. (2022), who introduced novel WL aggregation strategies that incorporate specific substructures like cycles and cliques. Moreover, Toenshoff et al. (2021) employed random walk techniques to generate small substructures for GNNs.

**Subgraph GNNs**. Recent approaches have explored breaking graph symmetry to better distinguish highly symmetric structures. These methods generate subgraphs according to predefined policies and then aggregate features across all subgraphs. Various subgraph generation strategies have been investigated. For instance, (Cotta et al., 2021) proposed a node deletion strategy, while (Bevilacqua et al., 2022) studied an edge deletion method. Additionally, node marking techniques have been explored by (Papp & Wattenhofer, 2022), and ego-network-based approaches have been examined in (Zhao et al., 2022; Zhang & Li, 2021; You et al., 2021). ID-GNN (You et al., 2021) extracts ego-networks and labels the root node to break symmetry, while NGNN (Zhang & Li, 2021) uses subgraph pooling to encode rooted subgraphs, improving expressiveness for distinguishing regular graphs. For the aggregation mechanisms of subgraphs, some methods have been explored to improve expressiveness by leveraging structural information. For instance, ESAN (Bevilacqua et al., 2022), performs layer-wise aggregation across subgraphs, enhancing expressive power. Building on this, Frasca et al. (2022) proposed a relaxed symmetry analysis, demonstrating that the expressiveness of such approaches is bounded by 3-WL. Furthermore, Qian et al. (2022) established connections between subgraph GNNs and $k$-FWL, introducing a learnable subgraph generation policy for improved flexibility.

**Multi-aggregation GNNs**. Several existing works have been developed to instantiate the $K$-hop or $K$-path message passing framework. For instance, MixHop (Abu-El-Haija et al., 2019) applies message passing on each hop using a graph diffusion kernel, then concatenates the representations across all hops to form the final output. K-hop sequentially executes message passing from hop $K$ to hop 1 to compute the central node's representation, though its computational procedure limits parallelizability in (Nikolentzos et al., 2020). MAGNA (Wang et al., 2021) incorporates an attention mechanism into $K$-hop message passing. Similarly, GPR-GNN (Chien et al., 2021) employs a graph diffusion kernel to perform graph convolution on $K$-hop neighborhoods and aggregates the results using learnable parameters. However, these approaches do not provide a formal definition of $K$-hop message passing or analyze its theoretical representational capabilities and limitations. Additionally, PathNNs (Michel et al., 2023) introduces a model that updates node representations by aggregating paths originating from the nodes. Path Integral-Based Convolution and Pooling for GNNs (Ma et al., 2020) proposes a

framework called PAN, which leverages path integrals to enhance GNN expressiveness and pooling mechanisms.

## B. The Weisfeiler-Lehman (WL) Algorithm and Its Variants

In this section, we describe the existing WL algorithm and its some variants.

### B.1. Weisfeiler-Lehman Algorithm

The 1-dimensional Weisfeiler-Lehman algorithm (1-WL), also known as the *color refinement* algorithm, iteratively computes a color mapping $\chi_G$ for a graph $G = (V, E)$, where each vertex $v \in V$ is assigned a color $\chi_G(v) \in \mathcal{C}$, with $\mathcal{C}$ denoting the set of colors. The pseudo-code for 1-WL is presented in Algorithm 1. Initially, all vertices are assigned the same color. At each iteration $l$, the algorithm updates the color of each vertex $v$ by combining its current color with the multiset of colors of its neighbors by a hash function, denoted as $m^l(v)$. Then the hash function is then applied to produce the new color $\chi^l(v)$. This procedure is repeated for a predefined number of iterations $L$ or until convergence.

---

**Algorithm 1** The 1-WL Algorithm

---

1: **Input:** Graph $G = (V, E)$ and the number of iterations $L$.
2: **Initialize:** Choose a fixed (arbitrary) element $c_0 \in \mathcal{C}$, and set $\chi^0(v) := c_0$ for all $v \in \mathcal{V}$.
3: **for** $l \leftarrow 1$ **to** $L$ **do**
4:     **for each** $v \in \mathcal{V}$ **do**
5:         $m^l(v) = Hash(\{\{\chi^{l-1}(u)|u \in N(v)\}\})$
6:         $\chi^l(v) := Hash\left(\chi^{l-1}(u), m^l(v)\right)$
7:     **end for**
8: **end for**
9: **Output:** Color mapping $\chi : \mathcal{V} \to \mathcal{C}$.

---

At each iteration, the color mapping induces a partition of the vertex set $V$. A key property of the 1-WL algorithm is that each iteration refines this partition, resulting in a progressively finer partition. Since the vertex set $V$ is finite, the algorithm is guaranteed to reach a stabilization point where the partition no longer changes.

The 1-WL algorithm can be used to determine whether two graphs $G$ and $H$ are isomorphic by comparing their representations. It is a fast and effective method for many practical applications, particularly when dealing with large graphs or when a quick approximation is sufficient. If the two representations differ, the graphs are not isomorphic, making 1-WL a necessary condition for graph isomorphism. However, the 1-WL test may fail to distinguish between two non-isomorphic graphs $G$ and $H$. This limitation has motivated the development of more expressive higher-order WL tests, as discussed in the next subsection.

### B.2. The $k$-dimensional Weisfeiler-Lehman Algorithm ($k$-WL)

The $k$-dimensional Weisfeiler-Lehman algorithm ($k$-WL) extends the 1-WL algorithm by enhancing its expressive power through the coloring of $k$-tuples of vertices from $V(G)^k$ instead of individual vertices.

To begin, we introduce the concept of a neighborhood in the context of $k$-WL. For a $k$-tuple $s = (s_1, \ldots, s_k) \in V(G)^k$, the $j$-th neighborhood $N_j(s)$ is defined as:

$$N_j(s) = \{(s_1, \ldots, s_{j-1}, r, s_{j+1}, \ldots, s_k) \mid r \in V(G)\}, \tag{18}$$

where $N_j(s)$ is formed by replacing the $j$-th component of $s$ with every vertex $r \in V(G)$. This definition enables the algorithm to explore the structural context of the $k$-tuple $s$ by systematically iterating over all possible substitutions for its $j$-th component.

This algorithm computes a coloring function $\chi^l : V(G)^k \to \mathcal{C}$ at each iteration $t$. At iteration $l = 0$, each $k$-tuple $s \in V(G)^k$ is assigned an initial label based on its *atomic type*. Specifically, two $k$-tuples $s$ and $s'$ receive the same color if the map $s_i \mapsto s'_i$ induces an isomorphism between the subgraphs induced by the nodes in $s$ and $s'$, respectively. For $l > 0$, the algorithm updates the label of each $k$-tuple $s$ by aggregating information from its $j$-th neighborhoods $N_j(s)$. The pruned

labels are computed as follows:

$$C_j^l(s) = Hash\big(\{\chi^{l-1}(s') \mid s' \in N_j(s)\}\big),\tag{19}$$

where $C_j^l(s)$ captures the aggregated information from the $j$-th neighborhood of $s$. The final label of $s$ at iteration $l$ is then computed as:

$$\chi^l(s) = Hash\big(\chi_l(s), (C_1^l(s), \dots, C_k^l(s))\big).\tag{20}$$

This refinement ensures that two tuples $s$ and $s'$ with identical labels at iteration $t-1$ will receive different labels at iteration $t$ if there exists a $j \in [1:k]$ such that the distribution of $j$-neighbors with specific colors differs between $s$ and $s'$. In summary, the pseudo-code for SC-WL is provided in Algorithm 2.

---

**Algorithm 2** The k-WL Algorithm

---

1: **Input:** Graph $G = (V, E)$ and the number of iterations $L$.
2: **Initialize:** Define tuples $V(G)^k$, choose a fixed element $c_0 \in \mathcal{C}$, and set $\chi^0(v) := c_0$ for all $s \in V(G)^k$.
3: **for** $l \leftarrow 1$ **to** $L$ **do**
4:     **for each** $s \in V(G)^k$ **do**
5:         $C_j^l(s) = Hash\big(\{\{\chi^{l-1}(s') \mid s' \in N_j(s)\}\}\big)$
6:         $\chi^l(s) = Hash\big(\chi_l(s), (C_1^l(s), \dots, C_k^l(s))\big)$
7:     **end for**
8: **end for**
9: **Output:** Color mapping $\chi : V(G)^k \to \mathcal{C}$.

---

By operating on $k$-tuples of vertices rather than individual nodes, the $k$-WL algorithm substantially enhances its capability to distinguish between non-isomorphic graphs. However, this improvement in expressive power comes at the expense of increased computational complexity, which grows rapidly as $k$ increases. Consequently, practical implementations typically involve a trade-off between the value of $k$ and the associated computational cost.

### B.3. The $K$-Path Weisfeiler-Lehman Algorithm ($K$-Path WL)

The $K$-Path WL algorithm extends the classical WL method by incorporating paths of length $K$, rather than focusing solely on the immediate neighborhood of vertices. To begin, we recall the definition of the $k$-path neighborhood. Let $N_{walk}^k(v)$ denote the set of all distinct paths of length $k$ starting from vertex $v$. A path $(v, u_1, u_2, \dots, u_k)$ is defined as a sequence of vertices and edges in which no edge or vertex is repeated. We say that a vertex $u_k$ is a $k$-path neighbor of $v$ if there exists a path $(v, u_1, \dots, u_k) \in N_{walk}^k(v)$, with $u_k$ being the terminal vertex of the path.

At the beginning, each vertex $v \in V$ is assigned an initial label $\chi^0(v)$, typically derived from vertex-specific features such as its degree or other attributes. In each iteration $l$, the algorithm updates the label $\chi_t^l(v)$ of each vertex by aggregating information from the $t$-path neighborhood for $t \in [1, K]$. Specifically, for each $t \in [1, K]$,

$$\chi_t^l(v) = Hash\left(\chi^{l-1}(v), \{\chi^{l-1}(u) \mid u \in N_{path}^t(v)\}\right).\tag{21}$$

Then, the algorithm updates the original label $\chi^l(v)$ by aggregating the labels from the $t$-path neighborhoods $\chi_t^l(v)$ for all $t \in [1, K]$. Specifically,

$$\chi^l(v) = Hash\left(\{\chi_t^l(v) \mid t \in [1, K]\}\right).\tag{22}$$

This process of refinement continues until the labels stabilize, meaning no further updates occur during an iteration. The pseudocode for the $K$-pWL algorithm is provided in Algorithm 3.

---

**Algorithm 3** The $K$-Path WL Algorithm

---

1: **Input:** Graph $G = (V, E)$, number of iterations $L$.
2: **Initialization:** $\forall v \in V, \chi^0(v), l = 0$.
3: **while** $l \leq L$ **do**
4:     $l = l + 1$.
5:     **for** $v \in V$ **do**
6:         **for** $t \in [1, K]$ **do**
7:             $\chi_t^l(v) = Hash(\{\{\chi^{l-1}(u) \mid u \in N_{path}^t(v)\}\})$.
8:         **end for**
9:         $\mathbf{X}^l(v) = (\chi_1^l(v), \chi_2^l(v), \cdots, \chi_K^l(v))$.
           $\chi^l(v) = Hash(\chi^{l-1}(v), \mathbf{X}^l(v))$.
10:     **end for**
11: **end while**
12: **Output:** Final labels $\chi^L(v)$ for all $v \in V$.

---

While this method is heuristic and may yield false positives or negatives, it provides a practical tool for many applications. This extension improves the algorithm's capacity to differentiate between large and intricate graphs, making it especially valuable for applications in graph databases and pattern recognition.

### B.4. The $K$-Hop Weisfeiler-Lehman Algorithm ($K$-Hop WL)

The $K$-Hop WL algorithm extends the classical WL method by considering the $K$-hop neighborhoods of vertices, rather than focusing solely on their immediate neighbors (Nikolentzos et al., 2020). To begin, we recall the definition of the $t$-hop neighborhood. Let $N_{hop}^t(v)$ denote the set of vertices that are exactly at a distance of $t$ hops from vertex $v$, where the distance is measured by the shortest path between $v$ and each vertex in $N_{hop}^t(v)$. It is important to note that the $t$-hop neighborhoods at different hop levels are disjoint, meaning that each vertex belongs to exactly one $t$-hop neighborhood for a given $t$.

At the start, each vertex $v \in V$ is assigned an initial label $\chi^0(v)$, typically derived from vertex-specific features such as its degree or other attributes. In each iteration $l$, the algorithm updates the label $\chi_t^l(v)$ of each vertex by aggregating information from its $t$-hop neighborhood for $t \in [1, K]$. Specifically, for each $t \in [1, K]$,

$$\chi_t^l(v) = Hash\left(\chi^{l-1}(v), \left\{\chi^{l-1}(u) \mid u \in N_{hop}^t(v)\right\}\right). \tag{23}$$

Then, the algorithm updates the original label $\chi^l(v)$ by aggregating the labels from the $t$-hop neighborhoods $\chi_t^l(v)$ for all $t \in [1, K]$. Specifically,

$$\chi^l(v) = Hash\left(\left\{\chi_t^l(v) \mid t \in [1, K]\right\}\right). \tag{24}$$

This refinement process continues until the labels stabilize, meaning no further changes occur in an iteration. The pseudocode for the $K$-hop WL algorithm is provided in Algorithm 4.

---

**Algorithm 4** The $K$-Hop WL Algorithm

---

1: **Input:** Graph $G = (V, E)$, number of iterations $L$.
2: **Initialization:** $\forall v \in V, \chi^0(v), l = 0$.
3: **while** $l \leq L$ **do**
4:     $l = l + 1$.
5:     **for** $v \in V$ **do**
6:         **for** $t \in [1, K]$ **do**
7:             $\chi_t^l(v) = Hash(\{\{\chi^{l-1}(u) : u \in N_{hop}^t(v)\}\})$.
8:         **end for**
9:         $\mathbf{X}^l(v) = (\chi_1^l(v), \chi_2^l(v), \cdots, \chi_K^l(v))$.
           $\chi^l(v) = Hash(\chi^l(v), \mathbf{X}^l(v))$.
10:     **end for**
11: **end while**
12: **Output:** Final labels $\chi(v)$ for all $v \in V$.

---

## C. The Pruned Weisfeiler-Lehman Algorithm and Its Variants

In this section, we present our pruned versions of the WL algorithms and provide an intuitive comparison between the standard WL and the pruned WL, highlighting their performance differences.

### C.1. The Pruned Weisfeiler-Lehman Algorithm (PR WL)

The Pruned Weisfeiler-Lehman (PWL) algorithm builds upon the standard WL method by introducing additional steps that refine the label update process using hierarchical hashing. Unlike the original WL algorithm, PWL employs a hierarchical approach for label refinement. Initially, each vertex $v$ is assigned an initial label $\chi^0(v)$. During the $l$-th iteration, each vertex aggregates the labels of its $2^{l-1}$-walk neighbors, specifically:

$$m_{2^{l-1}}^l(v) = Hash\left(\{\{\chi^{l-1}(u) : u \in N_{walk}^{2^{l-1}}(v)\}\}\right). \tag{25}$$

Subsequently, each vertex updates its label by applying a hash function to both its previous label and the aggregated neighborhood information, as follows:

$$\chi^l(v) = Hash\left(\chi^{l-1}(v), m_{2^{l-1}}^l(v)\right). \tag{26}$$

The pseudocode for the Pruned WL algorithm is provided in Algorithm 5.

---

**Algorithm 5** The PR WL Algorithm

1: **Input:** Graph $G = (V, E)$, number of iterations $L$.
2: **Initialization:** $\forall v \in V, \chi^0(v), l = 0$.
3: **while** $l \leq L$ **do**
4:    $l = l + 1, t = 1$.
5:    $m_1^l(v) = Hash(\{\{\chi^{l-1}(u) : u \in N(v)\}\})$.
6:    **for** $\forall v \in V$ **do**
7:       **while** $t < 2^{l-1}$ **do**
8:          $t = t + 1$.
9:          $m_t^l(v) = Hash(\{\{m_{t-1}^l(u) : u \in N(v)\}\})$.
10:       **end while**
11:       $\chi^l(v) = Hash(\chi^{l-1}(v), m_{2^{l-1}}^l(v))$.
12:    **end for**
13: **end while**
14: **Output:** Final labels $\chi^L(v)$ for all $v \in V$.

---

### C.2. The Pruned $K$-Path Weisfeiler-Lehman Algorithm(PR $K$-Path WL)

To address the computational overhead of the $K$-path WL algorithm, a pruned version is proposed, which selectively reduces the complexity by focusing on relevant paths at each iteration.

Unlike the standard $K$-path WL, the pruned version aggregates features from the $l$-path neighbors $N_{path}^l$ to the $K$-path neighbors $N_{path}^K$ at the $l^{th}$ layer ($l \leq K$), rather than from the 1-path neighbors $N_{path}^1$ to the $K$-path neighbors $N_{path}^K$. Let $\chi_v^l$ denote the feature of node $v$ at the $l^{th}$ layer. When $l \leq K$, the pruned $K$-pWL is defined as follows:

$$\chi_v^{l,t} = Hash(\chi^{l-1}(v), \{\{\chi^{l-1}(u)|u \in N_{path}^t(v)\}\})(l \leq t \leq K) \tag{27}$$

$$\chi_v^l = Hash(\{\{\chi^{l,k}(v)|k = l, \cdots K\}\}) \tag{28}$$

else when $l \geq K$, the $K$-path pruned Message passing framework can be defined as follows:

$$H_v^l = Hash(\chi^{l-1}(v), \{\{\chi^{l-1}(u)|u \in N_{path}^K(v)\}\}) \tag{29}$$

The pruned algorithm is outlined in Algorithm 6.

---

**Algorithm 6** The Pruned $K$-Path WL Algorithm

---

 1: **Input:** Graph $G = (V, E, X)$, number of iterations $L$.
 2: **Initialization:** $\forall v \in V$, $\chi^0(v)$, $l = 0$.
 3: **while** $l \leq K$ **do**
 4: $\quad$ $l = l + 1$.
 5: $\quad$ **for** $v \in V$ **do**
 6: $\quad\quad$ **for** $t \in [l, K]$ **do**
 7: $\quad\quad\quad$ $\chi_t^l(v) = Hash(\{\{\chi^{l-1}(u) : u \in N_{\text{path}}^t(v)\}\})$.
 8: $\quad\quad$ **end for**
 9: $\quad\quad$ $\mathbf{X}^l(v) = (\chi_l^l(v), \chi_{l+1}^l(v), \cdots, \chi_K^l(v))$.
$\quad\quad\quad$ $\chi^l(v) = Hash(\chi^{l-1}(v), \mathbf{X}^l(v))$.
10: $\quad$ **end for**
11: **end while**
12: **while** $l \leq L$ **do**
13: $\quad$ $l = l + 1$.
14: $\quad$ **for** $v \in V$ **do**
15: $\quad\quad$ $\chi^l(v) = Hash(\chi^{l-1}(v), \{\{\chi^{l-1}(u) : u \in N_{path}^L(v)\}\})$.
16: $\quad$ **end for**
17: **end while**
18: **Output:** Final labels $\chi^L(v)$ for all $v \in V$.

---

Compared to the standard $K$-Path WL algorithm, the pruned version achieves greater computational efficiency by dynamically adjusting the paths considered at each iteration. This reduction in complexity makes it more suitable for large-scale graphs while preserving much of the discriminative power of the original algorithm.

### C.3. The Pruned $K$-Hop Weisfeiler-Lehman Algorithm (PR $K$-Hop WL)

To further enhance computational efficiency, a pruned version of the $K$-hop WL algorithm is introduced. Unlike the standard $K$-hop WL, the pruned version aggregates features from the $l$-hop neighbors $N_{hop}^l$ to the $K$-hop neighbors $N_{hop}^K$ at the $l^{th}$ layer ($l \leq K$), rather than from the 1-hop neighbors $N_{hop}^1$ to the $K$-hop neighbors $N_{hop}^K$. Let $\chi_v^l$ denote the feature of node $v$ at the $l^{th}$ layer. When $l \leq K$, the pruned $K$-hWL is defined as follows:

$$\chi_v^{l,t} = Hash(\chi^{l-1}(v), \{\{\chi^{l-1}(u)|u \in N_{hop}^t(v)\}\})(l \leq t \leq K) \tag{30}$$

$$\chi_v^l = Hash(\{\{\chi^{l,k}(v)|k = l, \cdots K\}\}) \tag{31}$$

else when $l \geq K$, the $K$-Hop WL can be defined as follows:

$$H_v^l = Hash(\chi^{l-1}(v), \{\{\chi^{l-1}(u)|u \in N_{hop}^K(v)\}\}) \tag{32}$$

The pruned algorithm is outlined in Algorithm 7.

---

**Algorithm 7** The Pruned $K$-Hop WL Algorithm

---

1: **Input:** Graph $G = (V, E)$, number of iterations $L$.
2: **Initialization:** $\forall v \in V, \chi^0(v), l = 0$.
3: **while** $l \leq K$ **do**
4: $\quad l = l + 1$.
5: $\quad$ **for** $v \in V$ **do**
6: $\quad\quad$ **for** $t \in [l, K]$ **do**
7: $\quad\quad\quad \chi_t^l(v) = Hash(\{\{\chi^{l-1}(u) : u \in N_{\text{hop}}^t(v)\}\})$.
8: $\quad\quad$ **end for**
9: $\quad\quad \mathbf{X}^l(v) = (\chi_l^l(v), \chi_{l+1}^l(v), \cdots, \chi_K^l(v))$.
$\quad\quad \chi^l(v) = Hash(\chi^{l-1}(v), \mathbf{X}^l(v))$.
10: $\quad$ **end for**
11: **end while**
12: **while** $l \leq L$ **do**
13: $\quad l = l + 1$.
14: $\quad$ **for** $v \in V$ **do**
15: $\quad\quad \chi^l(v) = Hash(\chi^{l-1}(v), \{\{\chi^{l-1}(u) : u \in N_{hop}^L(v)\}\})$.
16: $\quad$ **end for**
17: **end while**
18: **Output:** Final labels $\chi^L(v)$ for all $v \in V$.

---

Compared to the standard $K$-Hop WL algorithm, the pruned version significantly reduces computational overhead by dynamically adjusting the neighborhoods considered during early iterations. This approach maintains much of the expressiveness of the original algorithm while being more scalable for large graphs.

### C.4. Process of WL Test, Pruned WL Test and Pruned 2-hop WL Test on Long-refinement Graph

Long-refinement graphs are a class of graphs for which the WL test does not terminate until it has iterated $|V| - 1$ times. As shown in Table 7, the adjacency lists of the long-refinement graph $G$ are provided. The Figures 4 and 5 illustrate the process of the WL test and the pruned WL test applied to graph $G$. During these processes, the WL test requires 11 iterations to terminate, whereas the pruned WL test and 2-Hop WL test only needs 4 iterations to achieve the same result as the WL test. Therefore, the pruned WL test and 2-Hop WL test is more efficient than the standard WL test.

| $v$ | $N(v)$ | $v$ | $N(v)$ |
|---|---|---|---|
| 0 | 1 | 6 | 3,7,8,9,11 |
| 1 | 0,2,3,4,5 | 7 | 2,6,8,9,10 |
| 2 | 1,3,5,7,10 | 8 | 5,6,7,10,11 |
| 3 | 1,2,4,6,10 | 9 | 4,6,7,10,11 |
| 4 | 1,3,5,9,11 | 10 | 2,3,7,8,9 |
| 5 | 1,2,4,8,11 | 11 | 4,5,6,8,9 |

*Table 7.* Adjacency lists of long-refinement graph $G$

The Figure 6 illustrates the process of the pruned WL test applied to graph $G$. The results illustrate the process of the WL test and the pruned 2-hop WL test applied to graph $G$. During these processes, the WL test requires 11 iterations of aggregation and combination to terminate, whereas the pruned 2-hop WL test only needs 11 iterations of aggregation and 4 iterations of combination to achieve the same result as the WL test. Therefore, the pruned 2-hop WL test is more efficient than the standard WL test.

## D. Detail of Matrix Language

Recently, Brijder (Brijder et al., 2019) and Geerts (Geerts, 2021) proposed a new matrix language called MATLANG. Matrix languages can be formalized as compositions of linear algebra operations. Intuitively, a linear algebra operation takes a

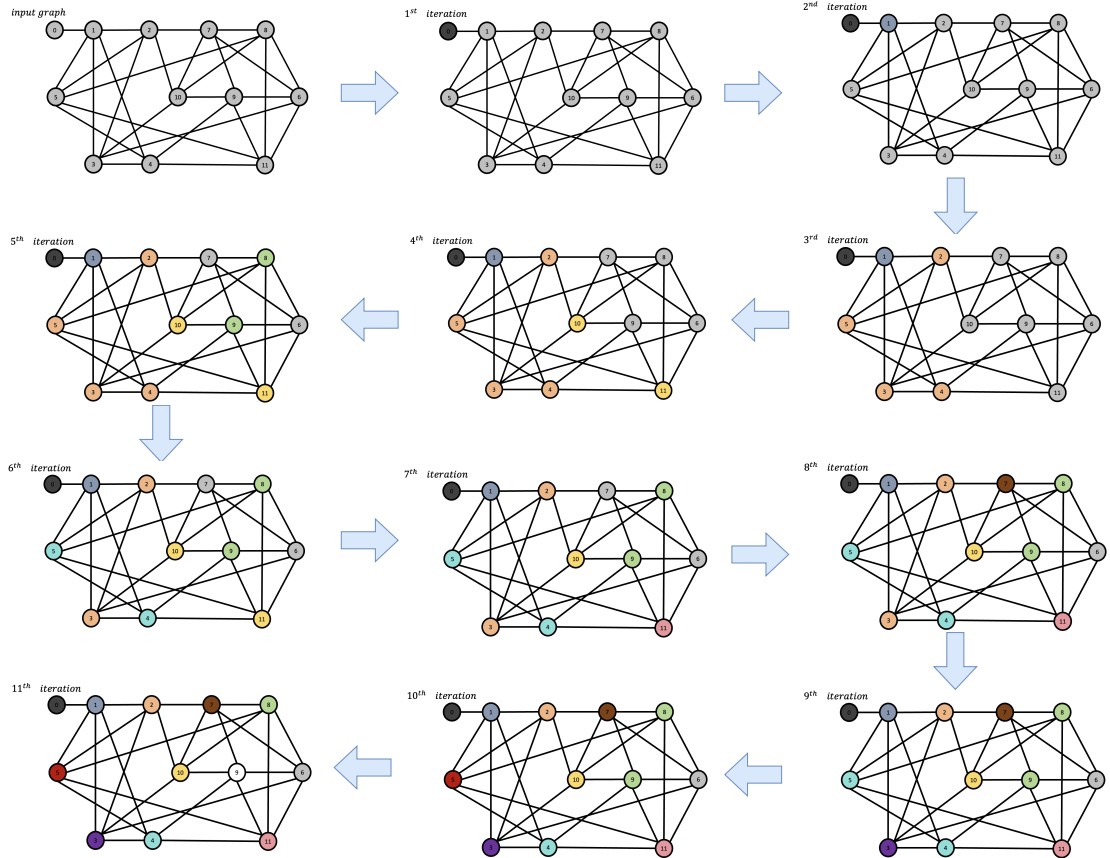

Figure 4. Process of WL test on Long-refinement Graph

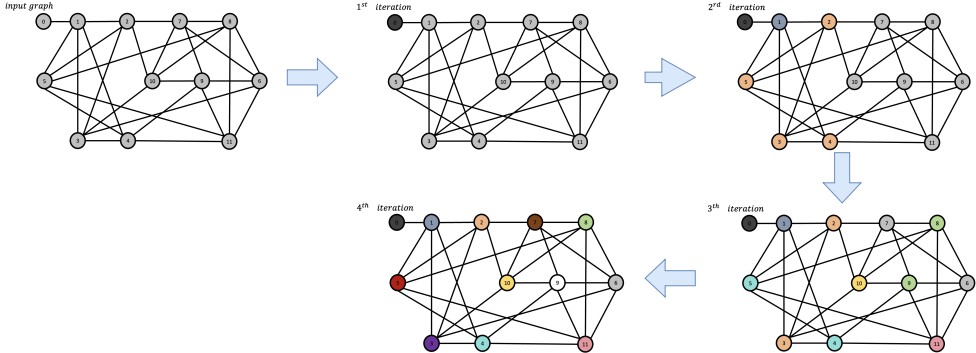

Figure 5. Process of pruned WL test on Long-refinement Graph

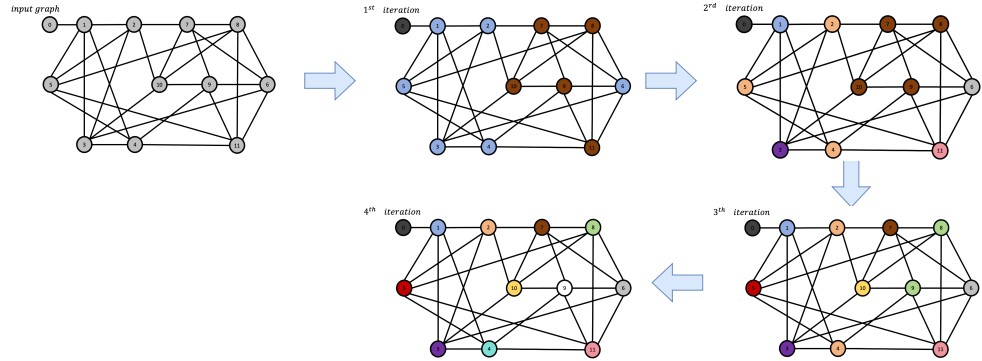

*Figure 6.* Process of pruned 2-hop WL test on Long-refinement Graph

number of matrices as input and returns another matrix (vector or scalar). More specifically, for linear algebra operations $op_1, \ldots, op_k$, the corresponding matrix query language is denoted by $ML(op_1, \ldots, op_k)$. This language includes various operations on matrices and establishes explicit connections between specific sets of operations and the WL tests, including the 1-WL and 3-WL tests. The expressive power of MATLANG varies depending on the operations included in each set.

**Definition D.1.** $ML(\mathcal{L})$ is a matrix language with an allowed set of operations $\mathcal{L} = \{op_1, \ldots, op_n\}$, where $op_i \in \{., +, ^\top, \mathrm{diag}, \mathrm{tr}, \mathbf{1}, \odot, \times, f\}$. The possible operations are introduced in the Appendix.

**Definition D.2.** A scalar sentence $e(X) \in \mathbb{R}$ in $ML^0(\mathcal{L})$ consists of any possible consecutive operations in $\mathcal{L}$ being applied to a given matrix $X$, resulting in a scalar value. Similarly, a vector sentence $e(X) \in \mathbb{R}^n$ in $ML^1(\mathcal{L})$ results in a vector, and a matrix sentence $e(X) \in \mathbb{R}^{n \times n}$ in $ML^2(\mathcal{L})$ results in a matrix.

For example, if $A$ is the adjacency matrix of a graph $G$, then $e(A) = \mathbf{1}^\top A \mathbf{1}$ is a scalar sentence in $ML^0(\mathcal{L})$ with $\mathcal{L} = \{., ^\top, \mathbf{1}\}$, computing the number of edges in $G$. Additionally, $e(A) = A\mathbf{1}$ is a vector sentence in $ML^1(\mathcal{L})$, computing the number of neighbors for each node in $G$. Geerts (Geerts, 2021) demonstrated that the languages $\mathcal{L}_1, \mathcal{L}_2,$ and $\mathcal{L}_3$ can characterize the WL test. The results are summarized as follows:

*Remark* D.3. Two adjacency matrices are indistinguishable by the 1-WL test if and only if $e(A_G) = e(A_{G'})$ for all $e \in ML^0(\mathcal{L}_1)$, where $\mathcal{L}_1 = \{., ^\top, \mathbf{1}, \mathrm{diag}\}$. Hence, all possible scalar sentences in $ML^0(\mathcal{L}_1)$ are identical for 1-WL equivalent adjacency matrices. Thus, $G \equiv_{1-WL} G' \leftrightarrow A_G \equiv_{ML^0(\mathcal{L}_1)} A_{G'}$. (See Theorem 7.1 in (Geerts, 2021).)

*Remark* D.4. Denote the operation $\odot_v$ as pointwise vector multiplication on vector sentences. The expressive power of $ML^0(\mathcal{L}_1)$ is equivalent to that of $ML^0(\mathcal{L}_5)$, where $\mathcal{L}_5 = \{., ^\top, \mathbf{1}, \odot_v\}$. Thus, for any two adjacency matrices $A_G$ and $A_{G'}$, they are indistinguishable by all possible scalar sentences in $ML^0(\mathcal{L}_1)$ if and only if they are indistinguishable by all possible scalar sentences in $ML^0(\mathcal{L}_5)$: $A_G \equiv_{ML^0(\mathcal{L}_5)} A_{G'} \leftrightarrow A_G \equiv_{ML^0(\mathcal{L}_1)} A_{G'}$. (see Proposition 8.1 in (Geerts, 2021))

*Remark* D.5. Two adjacency matrices are indistinguishable by the 3-WL test if and only if they are indistinguishable by any scalar sentence in $ML^0(\mathcal{L}_3)$, where $\mathcal{L}_3 = \{., ^\top, \mathbf{1}, \mathrm{diag}, \mathrm{tr}, \odot\}$. Thus, $G \equiv_{3-WL} G' \leftrightarrow A_G \equiv ML^0(\mathcal{L}_3)A_{G'}$. (see Theorem 9.2 in (Geerts, 2021))

*Remark* D.6. Enriching the operation set to $\mathcal{L}^+ = \mathcal{L} \cup \{+, \times, f\}$, where $\mathcal{L} \in \{\mathcal{L}_1, \mathcal{L}_2, \mathcal{L}_3\}$, does not improve the expressive power of the language. Thus, $A_G \equiv_{ML^0(\mathcal{L})} A_G \leftrightarrow A_G \equiv_{ML^0(\mathcal{L}^+)} A_G$. (see Proposition 7.5 in (Geerts, 2021))

For any two vectors $H_1, H_2 \in \mathbb{R}^n$, we denote $H_1 \sim_\sigma H_2$ if there exists a permutation $\sigma$ such that $\forall i \in [1, n], H_1(i) = H_2(\sigma(i))$. For any two matrices $A_1$ and $A_2$, we denote $A_1 \sim_\sigma A_2$ if there exists a permutation $\sigma$ such that $\forall i, j \in [1, n], A_1(i, j) = A_2(\sigma(i), \sigma(j))$. We denote $A_G \equiv_{ML^1(\mathcal{L})} A_{G'}$ if there exists a permutation $\sigma$ such that for all $e \in ML^1(\mathcal{L})$, $e(A_G) \sim_\sigma e(A_{G'})$. Similarly, we denote $A_G \equiv_{ML^2(\mathcal{L})} A_{G'}$ if there exists a permutation $\sigma$ such that for all $e \in ML^2(\mathcal{L})$, $e(A_G) \sim_\sigma e(A_{G'})$. In this research, we propose the language $ML(\mathcal{L}_4)$ with $\mathcal{L}_4 = \{., ^\top, \mathbf{1}, \mathrm{diag}, \odot\}$ and prove the equivalence between $ML(\mathcal{L}_4)$ and $ML(\mathcal{L}_3)$ as follows:

**Lemma D.7.** *For all $i \in \{0, 1, 2\}$, $A_G \equiv_{ML^i(\mathcal{L}_3)} A_G \leftrightarrow A_G \equiv_{ML^i(\mathcal{L}_4)} A_{G'}$.*

**Lemma D.8.** *For all $\mathcal{L} \in \{\mathcal{L}_1, \mathcal{L}_2, \mathcal{L}_3, \mathcal{L}_4\}$, $A_G \equiv_{ML^2(\mathcal{L})} A_{G'} \to A_G \equiv_{ML^1(\mathcal{L})} A_{G'} \to A_G \equiv_{ML^0(\mathcal{L})} A_{G'}$.*

Proofs are provided in the Appendix E.

The list of operations in Table 8 differs slightly from the list presented in Brijder Brijder et al., 2019: Instead of denoting

| | | |
|---|---|---|
| **conjugate transposition** ($\mathsf{op}(e) = e^*$) | | |
| $e(\nu(X)) = A \in \mathbb{C}^{m \times n}$ | $e(\nu(X))^* = A^* \in \mathbb{C}^{n \times m}$ | $(A^*)_{ij} = \overline{A}_{ji}$ |
| **one-vector** ($\mathsf{op}(e) = \mathbf{1}(e)$) | | |
| $e(\nu(X)) = A \in \mathbb{C}^{m \times n}$ | $\mathbf{1}(e(\nu(X)) = \mathbf{1} \in \mathbb{C}^{m \times 1}$ | $\mathbf{1}_i = 1$ |
| **diagonalization of a vector** ($\mathsf{op}(e) = \mathsf{diag}(e)$) | | |
| $e(\nu(X)) = A \in \mathbb{C}^{m \times 1}$ | $\mathsf{diag}(e(\nu(X)) = \mathsf{diag}(A) \in \mathbb{C}^{m \times m}$ | $\mathsf{diag}(A)_{ii} = A_i,$ $\mathsf{diag}(A)_{ij} = 0, i \neq j$ |
| **matrix multiplication** ($\mathsf{op}(e_1, e_2) = e_1 \cdot e_2$) | | |
| $e_1(\nu(X)) = A \in \mathbb{C}^{m \times n}$ $e_2(\nu(X)) = B \in \mathbb{C}^{n \times o}$ | $e_1(\nu(X)) \cdot e_2(\nu(X)) = C \in \mathbb{C}^{m \times o}$ | $C_{ij} = \sum_{k=1}^{n} A_{ik} \times B_{kj}$ |
| **matrix addition** ($\mathsf{op}(e_1, e_2) = e_1 + e_2$) | | |
| $e_1(\nu(X)) = A \in \mathbb{C}^{m \times n}$ $e_2(\nu(X)) = B \in \mathbb{C}^{m \times n}$ | $e_1(\nu(X)) + e_2(\nu(X)) = C \in \mathbb{C}^{m \times n}$ | $C_{ij} = A_{ij} + B_{ij}$ |
| **scalar multiplication** ($\mathsf{op}(e) = c \times e, c \in \mathbb{C}$) | | |
| $e(\nu(X)) = A \in \mathbb{C}^{m \times n}$ | $c \times e(\nu(X)) = B \in \mathbb{C}^{m \times n}$ | $B_{ij} = c \times A_{ij}$ |
| **trace** ($\mathsf{op}(e) = \mathsf{tr}(e)$) | | |
| $e(\nu(X)) = A \in \mathbb{C}^{m \times m}$ | $\mathsf{tr}(e(\nu(X)) = c \in \mathbb{C}$ | $c = \sum_{i=1}^{m} A_{ii}$ |
| **pointwise vector multiplication** ($\mathsf{op}(e_1, e_2) = e_1 \odot_v e_2$) | | |
| $e_1(\nu(X)) = A \in \mathbb{C}^{m \times 1}$ $e_2(\nu(X)) = B \in \mathbb{C}^{m \times 1}$ | $e_1(\nu(X)) \odot_v e_2(\nu(X)) = C \in \mathbb{C}^{m \times 1}$ | $C_i = A_i \times B_i$ |
| **pointwise matrix multiplication (Schur-Hadamard)** ($\mathsf{op}(e_1, e_2) = e_1 \odot e_2$) | | |
| $e_1(\nu(X)) = A \in \mathbb{C}^{m \times n}$ $e_2(\nu(X)) = B \in \mathbb{C}^{m \times n}$ | $e_1(\nu(X)) \odot e_2(\nu(X)) = C \in \mathbb{C}^{m \times n}$ | $C_{ij} = A_{ij} \times B_{ij}$ |
| **pointwise function application** ($\mathsf{op}(e_1, \ldots, e_p) = \mathsf{apply}[f](e_1, \ldots, e_p)$), $f : \mathbb{C}^p \to \mathbb{C} \in \Omega$ | | |
| $e_1(\nu(X)) = A^{(1)} \in \mathbb{C}^{m \times n}$ $\vdots$ $e_p(\nu(X)) = A^{(p)} \in \mathbb{C}^{m \times n}$ | $\mathsf{apply}[f]\big(e_1(\nu(X)), \ldots, e_p(\nu(X))\big) = B \in \mathbb{C}^{m \times n}$ | $B_{ij} = f(A_{ij}^{(1)}, \ldots, A_{ij}^{(p)})$ |

*Table 8.* Linear algebra operations supported in MATLANG and their semantics. In the last column, $^-$, $+$ and $\times$ denote complex conjugation, addition and multiplication in $\mathbb{C}$, respectively. Brijder et al., 2019

$\mathsf{ML}(\mathsf{op}_1, \ldots, \mathsf{op}_k)$ as every matrix language which returns vector, we denote $ML^2(\mathsf{op}_1, \ldots, \mathsf{op}_k)$ or $ML^1(\mathsf{op}_1, \ldots, \mathsf{op}_k)$ as every matrix language which returns another matrix (vector or scalar).

## E. Materials Of Proofs

### E.1. Proof of lemma D.7

**Lemma E.1.** *For all $i \in \{0, 1, 2\}$, $A_G \equiv_{ML^i(\mathcal{L}_3)} A_G \leftrightarrow A_G \equiv_{ML^i(\mathcal{L}_4)} A_{G'}$.*

*Proof.* Note that equation

$$e(A) = \mathbf{1}^T \cdot (A \odot diag(\mathbf{1})) \cdot \mathbf{1} = tr(A). \tag{33}$$

Hence we have $e \in ML^0(\mathcal{L}_4)$ that $e(A) = tr(A)$, which implies $\forall i \in \{0, 1, 2\}, ML^i(\mathcal{L})_3 = ML^i(\mathcal{L}_4)$.

$\square$

### E.2. Proof of Lemma D.8

**Lemma E.2.** *For all $\mathcal{L} \in \{\mathcal{L}_1, \mathcal{L}_2, \mathcal{L}_3, \mathcal{L}_4\}$, $A_G \equiv_{ML^2(\mathcal{L})} A_{G'} \to A_G \equiv_{ML^1(\mathcal{L})} A_{G'} \to A_G \equiv_{ML^0(\mathcal{L})} A_{G'}$.*

*Proof.* Since every vector or scalar value is generated by operation $\{., +, ^\top, diag, tr, \mathbf{1}, \odot, \times, f\}$. Hence if $\forall op \in \{., +, ^\top, diag, tr, \mathbf{1}, \odot, \times, f\}$, $op$ is permutation $\sigma$ preserved, then we can prove the lemma.

- Multiplication operation $\cdot$:

For all vectors $H_1, H_2, T_1, T_2 \in \mathbb{R}^n$, if $H_1 \sim_\sigma H_2$ and $T_1 \sim_\sigma T_2$, then

$$H_1 \cdot T_1 = \sum_i H_1(i) \cdot T_1(i) = \sum_i H_2(\sigma(i)) \cdot T_2(\sigma(i)) = \sum_i H_2(i) \cdot T_2(i) = H_2 \cdot T_2. \tag{34}$$

For all vectors $H_1, H_2 \in \mathbb{R}^n$ and matrices $A_1, A_2 \in \mathbb{R}^{n \times n}$, if $H_1 \sim_\sigma H_2$ and $A_1 \sim_\sigma A_2$, then

$$A_1 \cdot H_1(i) = \sum_j A_1(i,j) \cdot H_1(j) = \sum_j A_2(\sigma(i), \sigma(j)) \cdot H_2(\sigma(j)) = A_2 \cdot H_2(\sigma(i)). \tag{35}$$

For all matrices $A_1, A_2, B_1, B_2 \in \mathbb{R}^{n \times n}$, if $A_1 \sim_\sigma A_2$ and $B_1 \sim_\sigma B_2$, then

$$A_1 \cdot B_1(i,j) = \sum_k A_1(i,k) \cdot B_1(k,j) = \sum_k A_2(\sigma(i), \sigma(k)) \cdot B_2(\sigma(k), \sigma(j)) = A_2 \cdot B_2(\sigma(i), \sigma(j)). \tag{36}$$

Hence multiplication operation $\cdot$ is $\sigma$ preserved.

- Operation trace $tr$:

  For all matrices $A_1, A_2 \in \mathbb{R}^{n \times n}$, if $A_1 \sim_\sigma A_2$, then

$$tr(A_1) = \sum_i A_1(i,i) = \sum_i A_2(\sigma(i), \sigma(i)) = tr(A_2) \tag{37}$$

  by lemma $D.7$, there's the corresponding indicator vector of $tr$, $e(A) = (A \odot diag(\mathbf{1})) \cdot \mathbf{1}$, that $e(A_1) \sim_\sigma e(A_2) \rightarrow tr(A_1) = tr(A_2)$, hence the trace operation $tr$ is $\sigma$-preserved.

- Pointwise matrix (vector) multiplication (Schur-Hadamard) $\odot$:

  For all matrices $A_1, A_2, B_1, B_2 \in \mathbb{R}^{n \times n}$, if $A_1 \sim_\sigma A_2$ and $B_1 \sim_\sigma B_2$, then

$$A_1 \odot B_1(i,j) = A_1(i,j) \odot B_1(i,j) = A_2(\sigma(i), \sigma(j)) \odot B_2(\sigma(i), \sigma(j)) = A_2 \odot B_2(\sigma(i), \sigma(j)). \tag{38}$$

  For all vectors $H_1, H_2, T_1, T_2 \in \mathbb{R}^n$, if $H_1 \sim_\sigma H_2$ and $T_1 \sim_\sigma T_2$, then

$$H_1 \odot T_1(i) = H_1(i) \odot T_1(i) = H_2(\sigma(i)) \odot T_2(\sigma(i)) = H_2 \odot T_2(\sigma(i)). \tag{39}$$

  hence the pointwise matrix (vector) multiplication operation $\odot$ is $\sigma$-preserved.

- Operation of the one-vector $\mathbf{1}$: For all vectors $H_1, H_2 \in \mathbb{R}^n$, if $H_1 \sim_\sigma H_2$, then:

$$\mathbf{1}(i) = 1 = \mathbf{1}(\sigma(i)). \tag{40}$$

  Hence the operation of the one-vector $\mathbf{1}$ is $\sigma$-preserved.

It's obvious that operation $\{+, ^\top, \times, f\}$ is $\sigma$-preserved, hence $A_G \equiv_{ML^2(\mathcal{L})} A_{G'} \rightarrow A_G \equiv_{ML^1(\mathcal{L})} A_{G'} \rightarrow A_G \equiv_{ML^0(\mathcal{L})} A_{G'}$.

$\square$

### E.3. Proof of Lemma 4.2

**Lemma E.3.** *Given a positive integer sequence $a_k$, $S_k = \sum_{t \in [k]} a_t$, if $S_k > 2^k - 1$, then $a_k$ is not viewable.*

*Proof.* We first will prove $S_n \leq 2^n - 1$: when $n = 1$, if $a_n = 1$, then $S_n = 1 \leq 2^1 - 1$. If $a_n \neq 1$, then $S_n = 0 \leq 2^1 - 1$. Hence the lemma is true when $n = 1$.

Assume that the lemma is not true, hence there is a positive integer sequence $a_n^t$ and a positive integer $n_t$, where $a_n^{t'}$s length of view $S_n^t \geq 2^n$. Define a vector $X^{n_t} \in \{0,1\}^{n_t} (X_{n_t} \neq \{0\}^{n_t})$. We can establish a mapping $f : \{0,1\}^{n_t} \longrightarrow \mathbb{N}$:

$$f(X^{n_t}) = \sum_{1 \leq i \leq n_t, X_{n_t}(i)=1} a_i^t. \tag{41}$$

By the definition of the viewed integer sequence, $\forall 1 \leq l \leq S_n$, there exists an $X^{n_t} \in \{0,1\}^{n_t}(X_{n_t} \neq \{0\}^{n_t})$ such that $f(X^{n_t}) = l$. Since $S_n^t \geq 2^n$, the cardinality of the image of $f$: $|Im(f)| \geq 2^n$. However, on the other hand, the cardinality of $f$'s domain: $|\{0,1\}^{n_t}\backslash\{0\}^{n_t}| = 2^n - 1$, which induces a paradox because the cardinality of the image of the mapping is less than or equal to the cardinality of its domain. Hence, $\forall n \in \mathbb{N}_+, S_n \leq 2^n - 1$.

We now prove the equation holds iff $a_n = 2^{n-1}$: $\forall 1 \leq l \leq 2^n - 1$, $l$ has a corresponding vector $X^n \in \{0,1\}^n (X_n \neq \{0\}^n)$, such that $f(X) = l$, hence the $2^{n-1}$'s length view is $2^n - 1$. If there is another positive integer sequence $b_n$, its length view $S_n'$ is $2^n - 1$, $n_0$ is the smallest positive integer such that $a_{n_0} \neq b_{n_0}$, if $b_{n_0} \leq 2^{n_0-1} - 1$, then

$$\sum_{i \in \{1, \cdots, n_0\}} b_i \leq 2^{n_0} - 2. \tag{42}$$

Hence the length of the view of $b_n$, $S_{n_0}' \leq 2^{n_0} - 2$. If $b_{n_0} \geq 2^{n_0-1} + 1$, since

$$\sum_{i \in \{1, \cdots, n_0\}} b_i \geq 2^{n_0}, \tag{43}$$

if $S_{n_0}' = 2^{n_0} - 1$, the cardinality of the image of $f$: $|Im(f)| = 2^{n_0} - 1 + \left|\sum_{i \in \{1, \cdots, n_0\}} b_i\right| \geq 2^{n_0}$, but the cardinality of $f$'s domain: $|\{0,1\}^{n_0}\backslash\{0\}^{n_0}| = 2^{n_0} - 1$, which induces a paradox because the cardinality of the image of $f$ exceeds the cardinality of its domain. Hence the equation holds iff $a_n = 2^{n-1}$. □

### E.4. Proof of Theorem 4.1

**Theorem E.4.** *Given a positive integer sequence $a_k$ and a pair of graphs$(G, G')$, $S_k = \sum_{t \in [k]} a_t$, if $a_k$ is viewable, then $\forall l \in \mathbb{N}^+ \ GI_{a_k-walk}^l \subseteq GI_{WL}^{S_l}$.*

*Proof.* Since the sentences in $ML^1(\mathcal{L}_1)$ produce a scalar value which can be reached in the graph readout layer as a sum thanks to $\mathbf{1}^\top H^{(l_{end})}$, we need to show that if $a_k$ is viewable, the eq.8 can produce all possible vectors in $ML^1(\mathcal{L}_1)$ on the last node representation layer. Since every vector in $ML^1(\mathcal{L}_1)$ can be formed as:

$$A^k\mathbf{1}, Ae_i, diag(e_i)e_j(e_i, e_j \in ML^1(\mathcal{L}_1)). \tag{44}$$

We can define the degree of those vectors :

$$deg(A^k\mathbf{1}) = k, deg(A^k e_i) = k + deg(e_i), deg(diag(e_i)e_j) = max(deg(e_i), deg(e_j))(e_i, e_j \in ML^1(\mathcal{L}_1), k \in N). \tag{45}$$

Then we have the following conclusion:

**Lemma E.5.** *Given two adjacent matrices $A$ and $A$, $k \in N_+$, if there's permutation $\sigma$, $\forall 1 \leq t \leq k$, $A^t\mathbf{1} \sim_\sigma A'^t\mathbf{1}$, then $\forall e \in \{e|e \in ML^1(\mathcal{L}_1), deg(e) \leq k\}$, then $e(A) \sim_\sigma e(A')$.*

*Proof.* The lemma holds if $e = A^k\mathbf{1}$, since every vectors in $ML^1(\mathcal{L}_1)$ can be formed as:

$$A^k\mathbf{1}, Ae_i, diag(e_i)e_j(e_i, e_j \in ML^1(\mathcal{L}_1)). \tag{46}$$

Notice $diag(e_i)e_j$ is equivalent to $e_i \odot e_j$, therefore every vectors $e \in ML^1(\mathcal{L}_1)$ can be reformed as:

$$\prod_{i=1}^t \odot_v(A^{k_i}\mathbf{1}). \tag{47}$$

Where $k_i \in N^+$. By the definition of degree, $degree(\prod_{i=1}^m \odot(A^{k_i}\mathbf{1})) = max\{k_i|i \in [1,m]\}$. Let $k = max\{k_i|i \in [1,m]\}$. Then if $\forall 1 \leq t \leq k$, $A^t\mathbf{1} \sim_\sigma A'^t\mathbf{1}$, then $\forall 1 \leq q \leq n$

$$\prod_{i=1}^t A^{k_i}\mathbf{1}(q) = \prod_{i=1}^t A'^{k_i}\mathbf{1}(\sigma(q)). \tag{48}$$

Therefore,$\forall e \in \{e|e \in ML^1(\mathcal{L}_1), deg(e) \leq k\}$, $e(A) \sim_\sigma e(A')$. Thus, we have proved the lemma. □

**Lemma E.6.** *Given a pair of unlabeled graphs $(G, G')$, $A$ and $A'$ are their corresponding adjacency matrices, then if there's permutation $\sigma$, $\forall e \in ML^1(\mathcal{L}_1)$ that $deg(e) \leq k$, $e(A) \sim_\sigma e(A')$, then the $k^{th}$ feature $H_G^k \sim_\sigma H_{G'}^k$.*

*Proof.* Since the pair of unlabeled graphs are unlabeled, their nodes' feature can be represented as $\mathbf{1}$, hence when k=1, the feature is updated as following:

$$H^1 = MLP(\mathbf{1}, A \cdot \mathbf{1}). \tag{49}$$

Hence if there's permutation $\sigma$, $\forall e \in ML^1(\mathcal{L}_1)$ that $deg(e) \leq 1$, $e(A) \sim_\sigma e(A')$, then $H_G^1 \sim_\sigma H_{G'}^1$. Assume it holds when $k = k_0$, the feature is updated as following:

$$H^{k_0+1} = MLP(H^{k_0}, A \cdot H^{k_0}). \tag{50}$$

Since by assumption if $\forall e \in ML^1(\mathcal{L}_1)$ that $deg(e) \leq k_0$, $e(A) \sim_\sigma e(A')$, then we have $H_G^{k_0} \sim_\sigma H_{G'}^{k_0}$. Notice $\forall e \in ML^1(\mathcal{L}_1)$ that $deg(e) \leq k_0$, $deg(Ae) \leq k_0 + 1$, hence if $\forall e \in ML^1(\mathcal{L}_1)$ that $deg(e) \leq k_0 + 1$, $e(A) \sim_\sigma e(A')$, then $AH_G^{k_0} \sim_\sigma AH_{G'}^{k_0}$. Since MLP(WL test) can be regarded as apply$[f]$, by remark D.6, $H_G^{k_0+1} \sim_\sigma H_{G'}^{k_0+1}$. The lemma holds when $k = k_0 + 1$. $\square$

Now we can prove theorem 4.1: by lemma E.5 and lemma E.6, to prove $GI_{a_k-walk\,WL}^t \subseteq GI_{WL}^{S_t}$, we only need to prove that $\forall A^i \mathbf{1} (0 \leq i \leq S_t)$, there's $a_k - walk$ GNN with $t$ layers is able to generate:

When $t = 1$, if $a_1 \neq 1$ then $S_1 = 0$, in this case, $a_k - walk$ GNN is able to generate vector $\mathbf{1}$. if $a_1 \neq 1$ then $S_1 = 1$, in this case, $a_k - walk$ GNN is able to generate vector $\mathbf{1}$ and $A\mathbf{1}$. Hence it theorem 4.1 holds when $t1$.

Assume 4.1 holds when $t = t_0$. Then there's $a_k - walk$ GNN with $t_0$ layers is able to generate vector $A^i \mathbf{1} (0 \leq i \leq S_{t_0})$. By lemma 4.2, $\forall j$, $S_{t_0} + 1 \leq j \leq S_{t_0+1}$, there's subset of $N^+$: $L \subseteq \{1, \cdots, t_0, t_0 + 1\}$, where $t_0 + 1 \in L$, that $\sum_{l \in L} a_l = j$. Since $0 \leq j - a_{t_0+1} \leq S_{t_0}$, by assumption, there's $a_k - walk$ GNN with $t_0$ layers which is able to generate vector $A^{j-a_{t_0+1}} \mathbf{1}$. By eq.7

$$H^{t_0+1} = MLP(H^{t_0} W^{(t_0+1,1)} + A^{a_{t_0+1}} \cdot H^{t_0} W^{(t_0+1,2)}). \tag{51}$$

Hence there's $a_k - walk$ GNN with $t_0 + 1$ layers which is able to generate vector $A^{a_{t_0+1}} A^{j-a_{t_0+1}} \mathbf{1} = \mathbf{A^j 1}$. Hence $\forall A^i \mathbf{1} (0 \leq i \leq S_t)$, there's $a_k - walk$ GNN with $t$ layers is able to generate, therefore $GI_{a_k-walk\,WL}^t \subseteq GI_{WL}^{S_t}$. Now we have proved theorem 4.1. $\square$

### E.5. Proof of Theorem 4.3

**Theorem E.7.** *Given a positive integer sequence $a_k$, if $a_k$ is not viewable, then there's a pair of non-isomorphic graphs $(G, G')$ that $(G, G') \in GI_{a_k-walk\,WL}^l$ but $(G, G') \notin GI_{WL}^{S_t}$.*

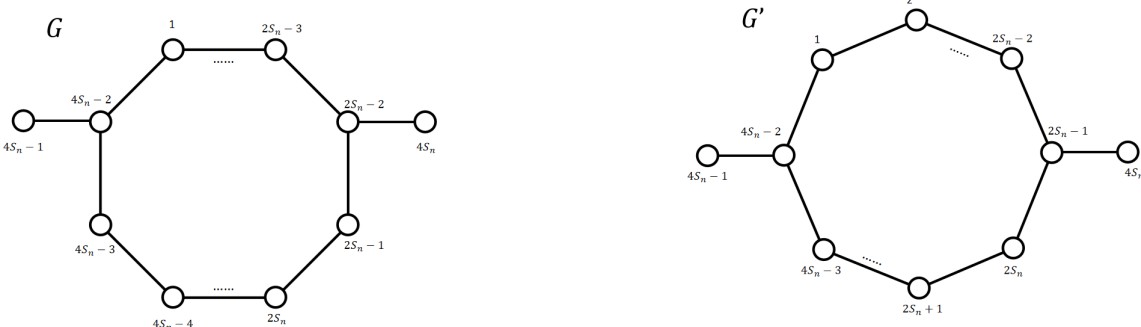

Figure 7. $(G, G') \in GI_{WL}^{S_n}$ while $(G, G') \notin GI_{a_n-walk\,WL}^n$

Given a positive integer sequence $a_n$, $S_n$ is the length view of $a_n$. If there's positive integer $S'_n > S_n$ that can be formed as $\sum_{t \in T} a_t$ where $T$ is subset of integer set $\{1, 2, \cdots, n\}$, observe that the counterexample graphs $G$ and $G'$ in figure 7, both $G$ and $G'$ have $4S'_n$ nodes, the edge set of $G$ is $E(G) = \{(i, i+1)|i \in [1, 4S'_n - 1]\} \cup \{(4S'_n - 2, 1)\} \cup \{(4S'_n, 2S'_n - 2)\}$ and the edge set of $G'$ is $E(G) = \{(i, i+1)|i \in [1, 4S'_n - 1]\} \cup \{(4S'_n - 2, 1)\} \cup \{(4S'_n, 2S'_n - 1)\}$.

**Lemma E.8.** $(G, G') \in GI_{WL}^{S_n}$ while $(G, G') \notin GI_{a_n-walk\ WL}^n$.

*Proof.* We first will prove $(G, G') \in GI_{WL}^{S_n}$: Let $label_{WL}^n(i)$ denotes the label node $i$ gets at $n^{th}$ iteration, We define the following permutation $\sigma$ mapping nodes from $G$ to $G'$ such as:

$$\sigma(i) = \begin{cases} i & if \quad i \in [1, S_n'] \cup [3S_n' - 1, 4S_n'] \\ i+1 & if \quad i \in [S_n' + 1, 3S_n' - 3] \\ S_n' + 1 & if \quad i = 3S_n' - 2 \end{cases} \tag{52}$$

In this way, we will prove that unravelling tree of node $i$ in $G$ and node $\sigma(i)$ in $G'$ at depth $S_n' - 1$ are isomorphic:

Let $dis(v, u)$ be the distance between node $v$ and $u$, the $k$-length distance induced graph of graph $G$, denoted as $G[i, k]$, whose node set is $V_{(i,k)} = \{u \in V | dis(v, u) \leq k\}$ and edge set is $E_{(i,k)} = \{\{u, w\} \in E | dis(v, u) \leq k, dis(v, w) \leq k\}$. Let $d_i = min(dis(i, 4S_n' - 1), dis(i, 4S_n')) - 1$, then as for node $i \in [1, 4S_n' - 2]/\{S_n', 3S_n' - 1, 3S_n' - 2, 3S_n' - 3\}$ in graph $G$, $i's$ $(S_n' - 1)$-length distance induced graph $G[i, S_n' - 1]$ is isomorphic to graph $H$ in figure which is a $2S_n'$-length path with an additional node adjacent to one of the nodes in path which is $d_i$-length distant from node $i$.

As for node $\sigma(i)'s (i \in [1, 4S_n' - 2]/\{S_n', 3S_n' - 1, 3S_n' - 2, 3S_n' - 3\})$ $(S_n' - 1)$-length distance induced graph in $G'$, $G'[\sigma(i), S_n' - 1]$ is isomorphic to $G[i, S_n' - 1]$, therefore their unravelling tree: $Unr_G^{S_n' - 1}(i)$ and $Unr_{G'}^{S_n' - 1}(\sigma(i))$ are isomorphic, node $i$ and $\sigma(i)$ get same label at $S_n' - 1^{th}$ iteration in WL test.

For node $i \in \{S_n', 3S_n' - 1, 3S_n' - 2, 3S_n' - 3\}$ in graph $G$, their $(S_n' - 1)$-length distance induced graphs: $G[i, S_n' - 1]$, are a $2S_n'$-length paths, while their corresponding node $\sigma(i)'s$ $(S_n' - 1)$-length distance induced graphs are also a $2S_n'$-length paths, therefore their unravelling tree: $Unr_G^{S_n' - 1}(i)$ and $Unr_{G'}^{S_n' - 1}(\sigma(i))$ are isomorphic, node $i$ and $\sigma(i)$ get same label at $S_n' - 1^{th}$ iteration in WL test.

For node $i \in \{4S_n' - 1, 4S_n'\}$ in graph $G$, their $(S_n' - 1)$-length distance induced graphs: $G[i, S_n' - 1]$, are a $2S_n' - 2$-length paths with additional node $i$ adjacent to the middle node in path. Also, their corresponding node $\sigma(i)'s$ $(S_n' - 1)$-length distance induced graphs are also are isomorphic to $G[i, S_n' - 1]$, therefore their unravelling tree: $Unr_G^{S_n' - 1}(i)$ and $Unr_{G'}^{S_n' - 1}(\sigma(i))$ are isomorphic, node $i$ and $\sigma(i)$ get same label at $S_n' - 1^{th}$ iteration in WL test.

In conclusion, every node $i \in [1, 4S_n]$, $i$ and $\sigma(i)$ get same label at $S_n' - 1^{th}$ iteration in WL test, therefore $(G, G') \in GI_{WL}^{S_n' - 1}$, since $S_n \leq S_n' - 1$, by the definition of WL test we have $(G, G') \in GI_{WL}^{S_n' - 1}$.

Next we will prove that $(G, G') \notin GI_{a_n-walk\ WL}^n$: in Wl test only node $S_n', 3S_n' - 1, 3S_n' - 2, 3S_n' - 3$ in graph $G$ and node $S_n', S_n' + 1, 3S_n' - 1, 3S_n' - 2$ in graph $G'$ gets the same label at iteration $S_n' - 1$. By theorem 4.1, $GI_{a_n-walk\ WL}^n \subseteq GI_{WL}^{S_n}$, hence if there's permutation $\sigma$(from $G$ to $G'$), then $\sigma$ will map $S_n'$ to $S_n', S_n' + 1, 3S_n' - 1\ or\ 3S_n' - 2$. On the other hand, node $S_n''s$ $S_n'$-walk neighbor $N_{walk}^{S_n'}(S_n')$ can be decomposed into two parts:$N_{path}^{S_n'}(S_n')$ and $N_{walk}^{S_n'}(S_n')/N_{path}^{S_n'}(S_n')$, while the second part is determined by $N_{walk}^l(l \leq S_n' - 1)$. Since $S_n'$ in graph $G$ has the same label as node $S_n', S_n' + 1, 3S_n' - 1, 3S_n' - 2$ in graph $G'$, their cardinality of second part $|N_{walk}^{S_n'}(S_n')/N_{path}^{S_n'}(S_n')|$ are equal, however cardinality of node $S_n''s$ first part equal to 4: $|N_{path}^{S_n'}(S_n')| = 4$ in graph $G$, while $S_n', S_n' + 1, 3S_n' - 1, 3S_n' - 2$ equal to 2 in graph $G'$, hence there's no node in graph $G'$ get the same label at $n^th$ iteration in $a_n$-walk WL test, hence $(G, G') \notin GI_{a_n-walk\ WL}^n$. $\square$

### E.6. Proof of Theorem 4.4

**Theorem E.9.** *Given a pair of graphs $(G, G')$ and $K \in \mathbb{N}^+$, then the expressiveness of pruned $K$-Path framework is as powerful as $K$-Path framework. In other word, $L \in \mathbb{N}^+ GI_{PRK-P}^L = GI_{K-P}^L$*

To prove theorem 4.4, we establish a connection with $ML^1(\mathcal{L}_1)$, we now extend the definition of matrix language $e \in ML^1(\mathcal{L}_1)$: let $P_{[K]}$ be the set of $k$-path matrices as $P_{[K]} = \{P_k | k \in [0, K]\}$, and vector sentence $e(P_{[K]})$ are formed by multiple operation on $e(X_{[K]})$ as follow:

$$e := \{X_1, \cdots, X_k\} | \mathsf{op}_1(e_1, \ldots, e_{p_1}) | \cdots | \mathsf{op}_k(e_1, \ldots, e_{p_k}) \tag{53}$$

where $\{X_1, \cdots, X_k\}$ denotes a set of matrix variables from an infinite set, serving as inputs to expressions, and $p_i$ denotes the number of inputs required by operation $\mathsf{op}_i$. Therefore $e(P_{[K]})$ denotes the comprehensive operation on $P_k$. For example:

let $e(P_{[2]}) = P_2\mathbf{1} + P_1\mathbf{1} = P_1^2\mathbf{1}$, then vector sentence $e(P_{[2]})$ denotes each node's the number of 2-length walks. We denote the equivalence between two path matrices set $P_{[K]}$ and $P'_{[K]}$ as $P_{[K]} \equiv_{ML^1(\mathcal{L}_1)} P'_{[K]}$, iff there's permutation $\sigma$, $\forall e \in ML^1(\mathcal{L}_1), e(P_{[K]}) \sim_\sigma e(P'_{[K]})$.

The proof is quite lengthy so we first describe its structure.

(1) First, we prove that $P_{[K]} \equiv_{ML^1(\mathcal{L}_1)} P'_{[K]}$ is equivalent to $G \equiv_{K-P} G'$.

(2) Second, we find the equivalent matrix language $ML(F^L_{(K,1)})$ for pruned $K$-Path framework, where $ML(F^L_{(K,K)})$ is equal to $ML^1(\mathcal{L}_1)$.

(3) Finally, we prove the equivalent expressive power for matrix language $ML(F^L_{(K,1)})$ and $ML(F^L_{(K,K)})$, therefore $K$-Path and its pruned frameworks have the same expressive power.

Regarding the connection between matrix language $ML^1(\mathcal{L}_1)$ and the expressive power of the $K$-Path framework, we have the following result:

**Theorem E.10.** $P_{[K]} \equiv_{ML^1(\mathcal{L}_1)} P'_{[K]} \leftrightarrow G \equiv_{K-P} G'$

*Proof.* To prove theorem E.10, we will prove any $label_i^l$ in $K$-path WL test will has a corresponding index vector sentence $e_i^l(P_{[K]}) \in ML^1(\mathcal{L}_3)$ that $e_i^l(P_{[K]}) = 1$ iff node $v$ gets $label_i$ at $l^{th}$ iteration.

Let $LABEL^l = \{label_i^l\}$ be the set of label nodes get at $l^{th}$ layer in $K$-path message passing framework's corresponding $K$-path WL test and $|LABEL^l| = \kappa_l$, hence if $label_i^l(v_i) = label_i^l(v_j)$ then node $v_i$ and $v_j$ will gets the same feature for any $l$-layers $K$-path message passing framework. If $\forall label_i^l$ there's corresponding sentence $e_i^l \in ml^1(\mathcal{L}_3)$ that $e_i^l(P_{[K]})(i) = 1$ if and only if node $v_i$ gets $label_i^l$ at $l^{th}$ iteration, then we can prove the theorem

First of all, since every node is not labeled, we denote the expression $e^{(0)}(X) := \mathbf{1}(X)$ as nodes' corresponding sentence. Next, when $l = L$, suppose by induction that we can have all $label_i^{L'}s$ corresponding sentence $e_i^L(P_{[K]})$, we will show that $ML^1(\mathcal{L}_3)$ is able to generate $label_i^{L+1'}s$ corresponding index vector sentence $e_i^{L+1}(P_{[K]})$. To this aim, we consider expressions:

$$m_{(i,k)}^{L+1}(P_{[K]}) = P_k \cdot e_i^L(P_{[K]}) \Big( 1 \le k \le K \Big). \tag{54}$$

The process of eq.54 is similar to the first equation in eq.6, $\forall r, q \in [1, n]$, we have $m_{(i,k)}^{L+1}(P_{[K]})(r) = m_{(i,k)}^{L+1}(P_{[K]})(q)$ iff $M_{v_r}^{L+1,k} = M_{v_q}^{L+1,k}$. Let $D_i^{L+1,k}$ be the set of values occurring in the column vector $m_{(i,k)}^{L+1}(P_{[K]})$, for $i = 1, 2, \cdots, \kappa_L$. We can compute, by means of an $ML^1(\mathcal{L}^+)$ expression, an indicator vector which identifies the rows for $M_v^{L+1,k}$ that hold a specific value $c \in D_i^{L+1,k}$. This expression is similar to the one used in the proof of theorem *cite*. More precisely, we consider expressions:

$$m_{=c}^{(L+1,i,k)}(P_{[K]}) = \left( \frac{1}{\prod_{c' \in D_i^{L+1,k}, c' \ne c}(c - c')} \right) \times \left( \prod_{c' \in D_i^{L+1,k}, c' \ne c} diag\Big( m_{(i,k)}^{L+1}(P_{[K]}) - c' \times \mathbf{1}(P_{[K]}) \Big) \right) \cdot \mathbf{1}(P_{[K]}). \tag{55}$$

for the current iteration $L + 1$ and value $c_i^k \in D_i^{L+1,k}$. Given these expressions, one can now easily obtain an indicator vector identifying all labels in $M_v^{L+1,k}$. Since $m_{=c_i^k}^{(L+1,i,k)}(P_{[K]})$ and $e_i^L(P_{[K]})$ are all $0 - 1$ vectors, to have an indicator vector of $H_v^{L+1,k}$, define the equation as follow:

$$m_{=c_i^k,j}^{(L+1,i,k)}(P_{[K]}) = diag(m_{=c_i^k}^{(L+1,i,k)}(P_{[K]})) \cdot e_j^L(P_{[K]})\Big( c_i^k \in D_i^{L+1,k}, j \in [1, \kappa_L] \Big). \tag{56}$$

Then $m_{=c_i^k,j}^{(L+1,i,k)}\Big( P_{[K]} \Big)(r) = 1$ iff in $K$-path WL test node $v_r$ get $label_{k,j}^{L+1}$ at $L + 1$ layer for $k$-path feature. Hence to have an indicator vector of $label^{L+1}$, we define the equation as follow:

$$m_{=c_i^1,\cdots c_i^K,j}^{(L+1,i)}(P_{[K]}) = \Big( \prod_{k\in[1,K]} diag(m_{=c_i^k,j}^{(L+1,i,k)}(P_{[K]})) \Big) \cdot \mathbf{1}(P_{[K]}) \Big( c_i^k \in D_i^{L+1,k}, j \in [1,\kappa_L] \Big). \tag{57}$$

Hence if $m_{=c_i^1,\cdots c_i^K,j}^{(L+1,i)}(P_{[K]})(r) = m_{=c_i^1,\cdots c_i^K,j}^{(L+1,i)}(P_{[K]})(q) = 1$ iff node $v_r$ and $v_q$ get the same label at iteration $L+1$. Hence let $W^{L+1} = |\{m_{=c_i^1,\cdots c_i^K,j}^{(L+1,i)}(P_{[K]})|c_i^k \in D_i^{L+1,k}, j \in [1,\kappa_L]\}|$ and $\kappa_{L+1} = |W^{L+1}|$, then we have already construct the $L+1^{th}$ layer's the set of vector indicator $e_i^{L+1}(1 \le i \le \kappa_{L+1})$. Since every indicator $e_i^{L+1} \in ML^1(\mathcal{L}_1)$, if there's permutation $\sigma$, $\forall e \in ML^1(\mathcal{L}_1)$, $e(P_{[K]}) \sim_\sigma e(P'_{[K]})$, then $\forall l \in N^+$ and $i \in \kappa_l$, $\{label_i^l(v)|v \in V\} = \{label_i^l(v')|v' \in V'\}$, therefore $G \equiv_{K-hop} G'$. Now we have proved the $P_{[K]} \equiv_{ML^1(\mathcal{L}_1)} P'_{[K]} \to G \equiv_{K-path} G'$.

To prove $P_{[K]} \equiv_{ML^1(\mathcal{L}_1)} P'_{[K]} \leftarrow G \equiv_{K-path} G'$, we can derive the proposition from the proof of theorem 4.1 as following:

**Proposition E.11.** $\forall e \in M^1(\mathcal{L}_1)$, $e(P_{[K]})$ has a equivalent form as

$$e(P_{[K]}) = \Big( \prod_{(i_1,\cdots i_K)\in N_+^K} diag\Big( (\prod_{k\in[1,K]} P_k^{i_k})\cdot\mathbf{1} \Big) \Big)\cdot\mathbf{1} \tag{58}$$

Therefore since the pair of unlabeled graphs are unlabeled, their nodes' feature can be represented as $\mathbf{1}$, hence when l=1, the feature is updated as following:

$$H^{1,k} = MLP(\mathbf{1}, P_k\cdot\mathbf{1}) \tag{59}$$

hence if there's permutation $\sigma$, $\forall e \in ML^1(\mathcal{L}_1)$ that $deg(e) \le 1$, $e(P_{[K]}) \sim_\sigma e(P'_{[K]})$, then $H_G^1 \sim_\sigma H_{G'}^1$. Assume it holds when $l = L$, the feature is updated as following:

$$H^{L+1,k} = MLP(H^{L,k}, P_k\cdot H^{L,k}) \tag{60}$$

since by assumption if $\forall e \in ML^1(\mathcal{L}_1)$ that $deg(e) \le L$, $e(P_{[K]}) \sim_\sigma e(P'_{[K]})$, then we have $H_G^L \sim_\sigma H_{G'}^L$. Notice $\forall e \in ML^1(\mathcal{L}_1)$ that $deg(e) \le k_0$, then $deg(P_k\cdot e) \le L+1$, hence if $\forall e \in ML^1(\mathcal{L}_1)$ that $deg(e) \le L+1$, $e(A) \sim_\sigma e(A')$, then $P_k\cdot H_G^L \sim_\sigma P_k'\cdot H_{G'}^L$. Since MLP(WL test) can be regarded as apply$[f]$, by remark D.6, $H_G^{L+1,k} \sim_\sigma H_{G'}^{L+1,k}$, therefore $H_G^{L+1} \sim_\sigma H_{G'}^{L+1}$. The proposition holds when $l = L+1$, now we have proved $P_{[K]} \equiv_{ML^1(\mathcal{L}_1)} P'_{[K]} \to G \equiv_{K-path} G'$

$\square$

To understand why this holds, consider the relationship between node configurations and the first layer of message passing. In the first layer of $K$-Hop message passing, each node aggregates information from its 1-path neighbors to its $K$-path neighbors. Since we assume no edge features and identical node features across the graph, the GNNs will only consider the local structural information of nodes, which corresponds to $P_k\mathbf{1}(1 \le k \le K)$. After the first iteration, all this information is encoded into node features and aggregated through different types of neighbors. Hence $P_{[K]} \equiv_{ML^1(\mathcal{L}_1)} P'_{[K]}$ implies that all the structural information extracted by the $K$-Path message passing framework is equivalent, leading to $G \equiv_{K-path} G'$.

We now present the equivalence in expressive power between the pruned $K$-Path framework and the original $K$-Path framework. By Theorem E.10, to prove this equivalence, we need to show that $G \equiv_{RE\ K-path} G' \to P_{[K]} \equiv_{ML^1(\mathcal{L}_1)} P'_{[K]}$.

Ivan (Jokić & Van Mieghem, 2022) has proved that k-path matrix $P_k$ can be generated by $\mathcal{L}_4 = \{., ^\top, \mathbf{1}, diag, \odot\}$, the conclusion can be stated as follow:

**Theorem E.12** ((Jokić & Van Mieghem, 2022)). $\forall k \in N^+$, $A$ is the adjacent matrix in graph $G$,there's $e_k \in ML^2(\mathcal{L}_4)$, that $e_k(A) = P_k$.

By theorem E.10, to prove theorem 4.4, we only need to prove that $G \equiv_{PR\ K-P} G' \to P_{[K]} \equiv_{ML^1(\mathcal{L}_1)} P'_{[K]}$, hence we have following conclusion:

**Lemma E.13.** *Let $P_{[K]}$ and $P'_{[K]}$ be two set of $k-$path matrices$(1 \leq k \leq K)$, if there's a set of vector sentences $E_1 \subset ML^1(\mathcal{L}_1) = \{.,^\top, \mathbf{1}, diag\}$ and vector sentence $e_1 \in ML^1(\mathcal{L}_1) = \{.,^\top, \mathbf{1}, diag\}$, if there's permutation $\sigma$, $\forall e' \in E_1 \, e'(P_{[K]}) \sim_\sigma e'(P'_{[K]})$ will imply $e_1(P_{[K]}) \sim_\sigma e_1(P'_{[K]})$, then $\forall e_2 \in ML^2(\mathcal{L}_1)$, if there's permutation $\sigma$, $\forall e' \in E_1 \left( e_2(P_{[K]}) \cdot e'(P_{[K]}) \right) \sim_\sigma \left( e_2(P'_{[K]}) \cdot e'(P'_{[K]}) \right)$ will imply $\left( e_2 \cdot (P_{[K]}) e_1(P_{[K]}) \right) \sim_\sigma \left( e_2(P'_{[K]}) \cdot e_1(P'_{[K]}) \right)$*

*Proof.* By the condition, there's function $f$, $e_1 = f(e'_1, \cdots, e'_s)$,$(|E_1| = s, e_i \in E_1)$, hence $\forall e_2 \in ML^2(\mathcal{L}_1)$, $e_2 \cdot e_1 = e_2 \cdot f(e'_1, \cdots, e'_s)$, therefore $\forall P_{[K]}, P'_{[K]}$, if there's permutation $\sigma$, $\forall e' \in E_1, \left( e_2(P_{[K]}) \cdot e'_1(P_{[K]}) \right) \sim_\sigma \left( e_2(P'_{[K]}) \cdot e'_1(P'_{[K]}) \right)$ then $\left( e_2(P_{[K]}) \cdot e_1(P_{[K]}) \right) \sim_\sigma \left( e_2(P'_{[K]}) \cdot e_1(P'_{[K]}) \right)$. $\qquad \square$

Lemma E.13 for example, since $P_1^2 \cdot \mathbf{1} = P_2 \cdot \mathbf{1} + P_1 \cdot \mathbf{1}$, hence if there's permutation $\sigma$, $\left( P_2 \cdot \mathbf{1} \right) \sim_\sigma \left( P'_2 \cdot \mathbf{1} \right)$ and $\left( P_1 \cdot \mathbf{1} \right) \sim_\sigma \left( P'_1 \cdot \mathbf{1} \right)$ hold will imply that $\left( P_1^2 \cdot \mathbf{1} \right) \sim_\sigma \left( P'^2_1 \cdot \mathbf{1} \right)$, hence if there's permutation $\sigma$, $\left( P_2^{l+1} \cdot \mathbf{1} \right) \sim_\sigma \left( P'^{l+1}_2 \cdot \mathbf{1} \right)$ and $\left( P_2^l \cdot P_1 \cdot \mathbf{1} \right) \sim_\sigma \left( P''^l_2 \cdot P'_1 \cdot \mathbf{1} \right)$ can deduce that $\left( P_2^l \cdot P'^2_1 \cdot \mathbf{1} \right) \sim_\sigma \left( P''^l_2 \cdot P'^2_1 \cdot \mathbf{1} \right)$.

Same as theorem 4.1, We can define the degree of those vectors sentence:

$$deg(\mathbf{1}) = 0, deg(P_i \cdot e) = 1 + deg(e), deg(diag(e_i)e_j) = max(deg(e_i), deg(e_j))(e_i, e_j \in \mathcal{L}_1, i \in [1, K]) \qquad (61)$$

Since by theorem E.10, $P_{[K]}$ and $P'_{[K]}$ are the set of $k-$path matrices of graph $G$ and $G'$, $P_{[K]} \equiv^{deg \leq l}_{\mathcal{L}_1} P'_{[K]}$ implies $G \equiv^l_{K-hop} G'$, hence to prove the equivalent expressive power of $K$-hop framework and $K$-hop refined framework, we only need to prove that $K$-hop refined framework is able to generate any vector sentence $e(P'_{[K]}) \in ML^1(\mathcal{L}_1) = \{.,^\top, \mathbf{1}, diag\}(deg(e) \leq l)$.

For $K$-hop framework and $K$-hop refined framework, we define the multiplicative structure of $k$-path-matrices set $P_{[K]}$, denoted as $F^L_{(K,t)}$, with $L$ layers and $t$-corner erased$(1 \leq t \leq K)$:

$$
\begin{array}{cccccccl}
P_1 & P_2 & P_3 & \cdots & P_t & \cdots & P_K & P_0 & 1^{st} \, layer \\
& P_2 & P_3 & \cdots & P_t & \cdots & P_K & P_0 & 2^{nd} \, layer \\
& & P_3 & \cdots & P_t & \cdots & P_K & P_0 & 3^{th} \, layer \\
& & & \vdots & & \vdots & \vdots & & \\
& & & & P_t & \cdots & P_K & P_0 & t^{th} \, layer \\
& & & & \vdots & & \vdots & \vdots & \\
& & & & P_t & \cdots & P_K & P_0 & L^{th} \, layer
\end{array}
\qquad (62)
$$

As 62 shows, when $l \leq t$, at $l^{th}$ layer, there are path-matrices from $l$-path to $K$-path with extra 0-path matrix(equal to identity matrix $I$). when $l \geq t$, at $l^{th}$ layer, there are path-matrices from $t$-path to $K$-path with extra 0-path matrix(equal to identity matrix $I$).

We define vector sentence $e(P_{[K]})$ as follow: pick up one arbitrary path matrix at every layer, calculate the product of them, then multiply the result with $\mathbf{1}$. We denote the set of those vector sentences and the pointwise multiplication of those vector sentences $\prod \odot_v e_i(P_{[K]})$ in $F^L_{K,t}$ as $ML(F^L_{(K,t)})$. Therefore, given two graph $G$ and $G'$, denote $G \equiv_{ML(F^L_{(K,t)})} G'$, if they are equivalent for any sentence in $ML(F^L_{(K,t)})(\forall e \in F^L_{(K,t)}$ there's permutation $\sigma, e(P_{[K]}) \sim_\sigma e(P'_{[K]}))$, we can conclude the following proposition from lemma E.13.

**Proposition E.14.** *If $t, K_1, K_1 \in N^+$, $t \leq K_1 \leq K_2$, then if theorem: for any pair of graph $(G, G')$, $G \equiv_{ML(F^L_{(K_1,t)})} G'$ holds, then theorem: for any pair of graph $(G, G')$, $G \equiv_{ML(F^L_{(K_2,t)})} G'$ holds.*

Since if $t_1 \leq t_2$, then $ML(F_{K,t_2}^L) \subseteq ML(F_{K,t_1}^L)$. Hence, as suggested below, Proposition E.14 demonstrates that if the equivalence between $ML(F_{(K,t_1)}^L)$ and $ML(F_{(K,t_2)}^L)$ holds for $(t_1 \leq t_2)$, then the equivalence between $ML(F_{(K+1,t_1)}^L)$ and $ML(F_{(K+1,t_2)}^L)$ holds:

$$
\begin{array}{llllll}
P_1 & P_2 \cdots & P_{t_1} \cdots & P_{t_2} \cdots & P_K & P_0 \\
& P_2 \cdots & P_{t_1} \cdots & P_{t_2} \cdots & P_K & P_0 \\
& \vdots & \vdots & \vdots & \vdots & \\
& P_{t_1} \cdots & P_{t_2} \cdots & P_K & P_0 & \\
& \vdots & \vdots & \vdots & \vdots & \\
& P_{t_1} \cdots & P_{t_2} \cdots & P_K & P_0 & \\
& \vdots & \vdots & \vdots & \vdots & \\
& P_{t_1} \cdots & P_{t_2} \cdots & P_K & P_0 &
\end{array}
\equiv
\begin{array}{llllll}
P_1 & P_2 \cdots & P_{t_1} \cdots & P_{t_2} \cdots & P_K & P_0 & 1^{st}\ layer \\
& P_2 \cdots & P_{t_1} \cdots & P_{t_2} \cdots & P_K & P_0 & 2^{nd}\ layer \\
& & \vdots & \vdots & \vdots & \vdots & \\
& & P_{t_1} \cdots & P_{t_2} \cdots & P_K & P_0 & t_1^{th}\ layer \\
& & & \vdots & \vdots & \vdots & \\
& & & P_{t_2} \cdots & P_K & P_0 & t_2^{th}\ layer \\
& & & \vdots & \vdots & \vdots & \\
& & & P_{t_2} \cdots & P_K & P_0 & l^{th}\ layer
\end{array}
$$

then we can derive from as:

$$
\begin{array}{llllll}
P_1 & P_2 \cdots & P_{t_1} \cdots & P_{t_2} \cdots & P_{K+1} & P_0 \\
& P_2 \cdots & P_{t_1} \cdots & P_{t_2} \cdots & P_{K+1} & P_0 \\
& \vdots & \vdots & \vdots & \vdots & \\
& P_{t_1} \cdots & P_{t_2} \cdots & P_{K+1} & P_0 & \\
& \vdots & \vdots & \vdots & \vdots & \\
& P_{t_1} \cdots & P_{t_2} \cdots & P_{K+1} & P_0 & \\
& \vdots & \vdots & \vdots & \vdots & \\
& P_{t_1} \cdots & P_{t_2} \cdots & P_{K+1} & P_0 &
\end{array}
\equiv
\begin{array}{llllll}
P_1 & P_2 \cdots & P_{t_1} \cdots & P_{t_2} \cdots & P_{K+1} & P_0 & 1^{st}\ layer \\
& P_2 \cdots & P_{t_1} \cdots & P_{t_2} \cdots & P_{K+1} & P_0 & 2^{nd}\ layer \\
& & \vdots & \vdots & \vdots & \vdots & \\
& & P_{t_1} \cdots & P_{t_2} \cdots & P_{K+1} & P_0 & t_1^{th}\ layer \\
& & & \vdots & \vdots & \vdots & \\
& & & P_{t_2} \cdots & P_{K+1} & P_0 & t_2^{th}\ layer \\
& & & \vdots & \vdots & \vdots & \\
& & & P_{t_2} \cdots & P_{K+1} & P_0 & L^{th}\ layer
\end{array}
$$

Now we are proving the equivalence between $ML(F_{(K,1)}^L)$ and $ML(F_{(K,K)}^L)$. We first prove it when $K = 2$, for example. When $L \leq K - 1$, $ML(F_{(2,2)}^L) = ML(F_{(2,1)}^L)$, the equivalence holds.

When $L \neq K$, the structures of $ML(F_{(K,1)}^L)$ and $ML(F_{(K,K)}^L)$ are as follow:

$$
\begin{array}{lll}
P_1 & P_2 & P_0 \\
P_1 & P_2 & P_0 \\
\vdots & \vdots & \vdots \\
P_1 & P_2 & P_0
\end{array}
\equiv
\begin{array}{llll}
P_1 & P_2 & P_0 & 1^{st}\ layer \\
& P_2 & P_0 & 2^{nd}\ layer \\
& \vdots & \vdots & \\
& P_2 & P_0 & L^{th}\ layer
\end{array}
$$

Since $P_1^2 = P_2 + P_1$, hence $\forall i \in N^+$, $P_1^i \mathbf{1}$, we can descend order of $P_1^i$ by replacing $P_1^2$ with $P_2 + P_1$, hence $P_1^i \mathbf{1}$ is determined by $P_2^{j_1} \cdot P_1 \mathbf{1}$ and $P_2^{j_2} \mathbf{1}$ ($0 \leq j_1 \leq [\frac{i_1}{2} - 1], 0 \leq j_2 \leq [\frac{i_1}{2}]$). Therefore $\forall i_1, i_2 \in N^+$, $P_2^{i_2} P_1^{i_1} \mathbf{1}$ is is determined by $P_2^{i_2+j_1} \cdot P_1 \mathbf{1}$ and $P_2^{i_2+j_2} \mathbf{1}$ ($0 \leq j_1 \leq [\frac{i_1}{2} - 1], 0 \leq j_2 \leq [\frac{i_1}{2}]$). On the other hand, $i_2 + j_1 + 1 \leq i_2 + [\frac{i_1}{2} - 1] + 1 \leq i_1 + i_2$, $i_2 + j_2 \leq i_2 + [\frac{i_1}{2}] \leq i_1 + i_2$, Which means $\forall L \in N^+$, $e \in ML(F_{(2,1)}^L)$, there's subset $E_e \subseteq ML(F_{(2,2)}^L)$ that vector sentence $e$ is determined by $E_e$, now we have proved the equivalence between $ML(F_{(2,1)}^L)$ and $ML(F_{(2,2)}^L)$.

Now assume the equivalence between $ML(F^L_{(k,k-1)})$ and $ML(F^L_{(k,k)})$ holds when $k \leq K$. Therefore, by proposition $E.14$, the equivalence between $ML(F^L_{(K+1,K-1)})$ and $ML(F^L_{(K+1,K)})$ holds. Hence we only need to prove the equivalence between $ML(F^L_{(K+1,K)})$ and $ML(F^L_{(K+1,K+1)})$. Let $e$ be a sentence, and $\omega(e(P_{[k]}))$ denotes the corresponding walk to $e(P_{[k]})$, for example, if $e(P_{[k]}) = P_2$, then $\omega(e(P_{[k]}))$ denotes all the 2-length path in graph. We first consider $\omega(P^2_K)/\omega(P_{K+1} \cdot P_{K-1})$: $\omega(P^2_K)/\omega(P_{K+1} \cdot P_{K-1})$ is the subset of $2K$-length walks in graph but the $K+1^{th}$ node is recurring in the walk sequence, hence the number of these walks from every node is determined by $P_K \cdot P_i (0 \leq i \leq K-1)$, in the same way $\omega(P_{K+1} \cdot P_{K-1})/\omega(P^2_K)$ is the subset of $2K$-length walks in graph but the $K+2^{th}$ node is recurring in the walk sequence, hence the number of these walks from every node is determined by $P_{K+1} \cdot P_j (0 \leq j \leq K-2)$. Therefore $P^2_K \mathbf{1}$ is determined by $P_K P_i \mathbf{1}$ and $P_{K+1} P_i \mathbf{1} (0 \leq i \leq K-1)$, which means that there's subset $E_{(P^2_K \mathbf{1})} \subseteq ML(F^L_{(K+1,K+1)})$ that vector sentence $P^2_K \mathbf{1}$ is determined by $E_{(P^2_K \mathbf{1})}$.

Now we are able to prove the equivalence between $ML(F^L_{(K+1,K)})$ and $ML(F^L_{(K+1,K+1)})$:

$$
\begin{array}{cccccc}
P_1 & P_2 \cdots & P_K & P_{K+1} & P_0 & \\
 & P_2 \cdots & P_K & P_{K+1} & P_0 & \\
 & & \vdots & \vdots & \vdots & \\
 & & P_K & P_{K+1} & P_0 & \\
 & & P_K & P_{K+1} & P_0 & \\
 & & \vdots & \vdots & \vdots & \\
 & & P_K & P_{K+1} & P_0 &
\end{array}
\quad \equiv \quad
\begin{array}{cccccc}
P_1 & P_2 \cdots & P_K & P_{K+1} & P_0 & 1^{st} \ layer \\
 & P_2 \cdots & P_K & P_{K+1} & P_0 & 2^{nd} \ layer \\
 & & \vdots & \vdots & \vdots & \\
 & & P_K & P_{K+1} & P_0 & K^{th} \ layer \\
 & & & P_{K+1} & P_0 & K+1^{th} \ layer \\
 & & & \vdots & \vdots & \\
 & & & P_{K+1} & P_0 & L^{th} \ layer
\end{array}
$$

if $e \in F^L_{(K+1,K)}$, but $e \notin F^L_{(K+1,K+1)}$, then

- $P^2_K$ is not in $e$: let $i_{K+1}$ equal to the order of $P_{K+1}$ in $e$ and $e_{P_{K+1}}$ be $e$ with $P^{i_{K+1}}_{K+1}$ erased. Then $e_{P_{K+1}} \in F^{L-i_{K+1}}_{(K,K-1)}$, by assumption, $e_{P_{K+1}}$ is determined by $F^{L-i_{K+1}}_{(K,K)}$, since $P^2_K$ is not in $e$, $P^{i_{K+1}}_{K+1} \cdot e_{P_{K+1}} = e$ is determined by $F^{L-i_{K+1}}_{(K+1,K+1)}$.

- $P^2_K$ is in $e$: let $P^2_K \cdot e' = e$, since $P^2_K \mathbf{1}$ is determined by $P_K P_i \mathbf{1}$ and $P_{K+1} P_i \mathbf{1} (0 \leq i \leq K-1)$, we are able to descend the order of $P_K$, until $P^2_K$ is not in, therefore $e$ are determined by the subset of $F^L_{(K+1,K)}$ where $P^2_K$ is not in every vector sentence, from the conclusion above those vector sentences are determined by $F^L_{(K+1,K+1)}$, therefore we have proved the equivalence between $ML(F^L_{(K+1,K)})$ and $ML(F^L_{(K+1,K+1)})$.

Therefore, we can prove the equivalence between $ML(F^L_{(K+1,1)})$ and $ML(F^L_{(K+1,K+1)})$: since $\forall k \in N^+$, $ML(F^L_{(k,k-1)})$ and $ML(F^L_{(k,k)})$ are equivalent, hence by proposition $E.14$ $ML(F^L_{(K,k-1)})$ and $ML(F^L_{(K,k)})(1 \leq k \leq K)$ are equivalent, hence the expressive power of $ML(F^L_{(K,1)})$ and $ML(F^L_{(K,K)})(1 \leq k \leq K)$ are equivalent.

Now that we can prove theorem 4.4: by theorem $E.10$, $P_{[K]} \equiv_{ML(F^L_{(K,K)})} P'_{[K]} \leftrightarrow G \equiv_{K-P} G'$ while $P_{[K]} \equiv_{ML(F^L_{(K,1)})} P'_{[K]} \leftrightarrow G \equiv_{PRK-P} G'$. We already obtain the conclusion that the expressive power of $ML(F^L_{(K,1)})$ and $ML(F^L_{(K,K)})(1 \leq k \leq K)$ are equivalent, hence we have $G \equiv_{K-P} G' \leftrightarrow G \equiv_{PRK-P} G'$, indicating $L \in N^+$ $GI^L_{PRK-P} = GI^L_{K-P}$, we have proved the expressiveness of pruned $K$-Path framework is as powerful as $K$-Path framework.

### E.7. Proof of Theorem 4.5

**Theorem E.15.** $\forall L, K \in N^+$, $(RG \cap GI^L_{PR2-H}) \subseteq (RG \cap GI^L_{2-H})$ and $(SRG \cap GI^L_{REK-H}) \subseteq (SRG \cap GI^L_{K-H})$.

To prove theorem 4.5, we connect its equivalence with matrix language. We establish a connection with $ML(\mathcal{L}_1)$, let us denote the set of $k$-hop matrices as $O[K] = \{O_k | k \in [0, K]\}$. We then arrive at the following conclusion:

**Theorem E.16.** $O_{[K]} \equiv_{ML^1(\mathcal{L}_1)} O'_{[K]} \leftrightarrow G \equiv_{K-Hop} G'$

To proof if theorem $E.16$ is same as theorem $E.10$, just replace $P_{[K]}$ with $O_{[K]}$. Given a pair of regular graphs $(G, G')$ $O_{[K]}$ and $O'_{[K]}$ are their $K$-hop matrices set, by theorem $E.16$, if 2-hop WL test decides that they are isomorphic, then $O_{[2]} \equiv_{ML^1(\mathcal{L}_3^1)} O_{[2]}$. Since $(G, G')$ are regular graphs, each node's number of neighbors is equal, hence $\forall i \in N^+$, $O_1^i \mathbf{1} = \prod_{=1}^i \odot_v (O_1 \mathbf{1})$, so $O_1^i \mathbf{1}$ is decided by $O_1 \mathbf{1}$. Hence if $e(O_{[2]}) \notin ML(F_{(2,2)}^L)$, then $e(O_{[2]})$ is formed as the pointwise multiplication of vector as $O_1^{i_1} O_2^{i_2} \mathbf{1}(i_1 + i_2 \leq L)$, which is determined by $O_1 \mathbf{1}$ and $O_2^{i_2} \mathbf{1}$, since $O_1 \mathbf{1}, O_2^{i_2} \mathbf{1} \in ML(F_{(2,2)}^L)$, hence if $(G, G')$ is a pair of regular graphs, then $O_{[2]} \equiv_{ML(F_{(2,2)}^L)} O'_{[2]} \rightarrow O_{[2]} \equiv_{ML(F_{(2,1)}^L)} O'_{[2]}$.

Before prove the the equivalence of $O_{[K]}$ between $ML(F_{(K+1,1)}^L)$ and $ML(F_{(K+1,K+1)}^L)$ on strong regular graph, we first start with a lemma:

**Lemma E.17.** *Let $G$ be an $(n, r, a, c)$ strongly regular graph, if $c = 0$, then $G$ is isomorphic to $mK_{r+1}$ for some $m \neq 2$($K_t$ is Complete graph with nodes).*

*Proof.* If $c = 0$, then any two neighbours of a vertex $v$ must be adjacent, and so $a = r - 1$, hence the component containing any vertex must be a complete graph $K_{r+1}$ and hence $G$ is a disjoint union of complete graphs. $\square$

Since $SRG \subset RG$, we can derive the equivalence of $O_{[2]}$ between $ML(F_{(K+1,1)}^L)$ and $ML(F_{(K+1,K+1)}^L)$ from the conclusion above, now we prove the equivalence of $O_{[3]}$ between $ML(F_{(K+1,1)}^L)$ and $ML(F_{(K+1,K+1)}^L)$: for each node $v's$ 2-hop neighbor $N_{hop}^2$, if $u \in N_{hop}^2$, then $u$ and $v$ have $c$ common neighbors, hence for every $v's$ 2-hop neighbor node $u$, there are $c$ paths from $v$ to $u$. Hence $cO_2 \mathbf{1} = P_2 \mathbf{1} = P_1^2 \mathbf{1} - P_1 \mathbf{1} = r(r-1)\mathbf{1} = (P_1 \mathbf{1} - \mathbf{1}) \odot P_1 \mathbf{1}$.

Next we consider the walks in $\omega(P_2^2)$, if $v$ and $u$ have a walk in $\omega(P_2^2)$ then $v$ and $u$ cannot be adjacent, hence $P_2^2 \mathbf{1} = P_2 \mathbf{1} \odot P_2 \mathbf{1}$, therefore $P_2^2 \mathbf{1}$ is determined by $P_2 \mathbf{1}$, hence if $e(O_{[2]}) \notin ML(F_{(3,3)}^L)$, then $e(O_{[2]})$ is formed as the pointwise multiplication of vector as $O_1^{i_1} O_2^{i_2} O_3^{i_3} \mathbf{1}(i_1 + i_2 + i_3 \leq L)$, which is determined by $O_1 \mathbf{1}$, $O_2 \mathbf{1}$ and $O_3^{i_3} \mathbf{1}$, since $O_1 \mathbf{1}, O_2 \mathbf{1}, O_3^{i_3} \mathbf{1} \in ML(F_{(3,3)}^L)$, hence if $(G, G')$ is a pair of regular graphs, then $O_{[3]} \equiv_{ML(F_{(3,3)}^L)} O'_{[2]} \implies O_{[3]} \equiv_{ML(F_{(3,1)}^L)} O'_{[2]}$.

Now we prove the equivalence of $O_{[K]}$ between $ML(F_{(K+1,1)}^L)$ and $ML(F_{(K+1,K+1)}^L)$ when $K \neq 4$: if $O_3 \mathbf{1} \neq \mathbf{0}$, then there is nodes $v$ and $u$, $v$ and $u$ do not have common neighbor, hence $c = 0$, therefore by $E.17$, graph $G$ is isomorphic to $mK_{r+1}$, since $K$-hop pruned WL test is able to output $n$ and $r$, hence $K$-hop pruned WL test will decide $(G, G')$ are isomorphic iff they are isomorphic to $mK_{r+1}(m \neq 2)$.

On the other hand, if $O_3 \mathbf{1} = \mathbf{0}$, then $\forall k \in N^+, K \neq 3, O_k \mathbf{1} = \mathbf{0}$, hence the output of $K$-hop WL test and $K$-hop pruned WL test for $(G, G')$ are the same, now we have proved the theorem.

### E.8. Detail of GNNs' Redundant Structure Task

For MP-GNN $\mathcal{M}$, the 4 parameter configurations of $\mathcal{M}$ which is competent for the task is listed as follow:

| configurations 1 | configurations 2 | configurations 3 | configurations 4 |
|---|---|---|---|
| $H_v^1 = \sigma(H_v^0 + \sum H_u^0)$ | $H_v^1 = \sigma(H_v^0 + \sum H_u^0)$ | $H_v^1 = \sigma(H_v^0 + \sum H_u^0)$ | $H_v^1 = \sigma(\sum H_u^0)$ |
| $H_v^2 = \sigma(H_v^1 + \sum H_u^1)$ | $H_v^2 = \sigma(H_v^1 + \sum H_u^1)$ | $H_v^2 = \sigma(\sum H_u^1)$ | $H_v^2 = \sigma(H_v^1 + \sum H_u^1)$ |
| $H_v^3 = \sigma(H_v^2 + \sum H_u^2)$ | $H_v^3 = \sigma(\sum H_u^2)$ | $H_v^3 = \sigma(H_v^2 + \sum H_u^2)$ | $H_v^3 = \sigma(H_v^2 + \sum H_u^2)$ |

For pruned MP-GNN $\mathcal{M}'$, the 2 parameter configurations of $\mathcal{M}'$ which is competent for the task is listed as follow:

| configurations 1 | configurations 2 |
|---|---|
| $H_v^1 = \sigma(H_v^0 + \sum H_u^0)$ | $H_v^1 = \sigma(H_v^0 + \sum H_u^0)$ |
| $H_v^2 = \sigma(H_v^1 + \sum \sum H_u^1)$ | $H_v^2 = \sigma(\sum \sum H_u^1)$ |

We now prove that those 6 models are competent for the task: It is obvious for both models the configurations 1 is competent for the task. We will categorize the conditions to discuss the rest configurations can also accomplish the task.

(1) $v$ is marked: for MP-GNN, since graph is connected with at least 3 nodes, node $v$ has at least one 2-hop neighbor $u$, its $H_u$ will be equal to 1 in two iterations such that $W_1^l \neq 0$, therefore $H_v^3 = 1$. As for pruned MP-GNN, since $H_v^1 = 1$, $H_v^2 = (A^2)_v \cdot H_v^1 = 1$.

(2) $v's$ 1-hop neighbor $u$ is marked: its $H_u$ will be equal to 1 in two iterations that $W_1^l \neq 0$, therefore $H_v^3 = 1$. As for pruned MP-GNN, we have $H_v^1 = 1$, hence $H_v^2 = 1$.

(3) $v's$ 2-hop neighbor $u$ is marked: its $H_u$ will be equal to 1 in two iterations that $W_1^i = 0$ and $W_1^l = 1$, therefore $H_v^3 = 1$. As for pruned MP-GNN, we have $H_u^1 = 1$, hence $H_v^2 = 1$.

(3) $v's$ 3-hop neighbor $u$ is marked: it's obvious since for both model will gather 3-hop-distance node information.

Now that if task outputs 1, iff at least one situation happens, therefore the rest configurations can also accomplish the task.

## F. Time and Space Complexity

In this section, we discuss more about the time and space complexity of MP, $K$-Path, $K$-Hop and their pruned frameworks, We suppose a graph has $n$ nodes, every framework is designed to gather node's information at a distance of $L$, $(n \gg L \gg K)$.

**Message Passing Framework:**

For WL test(MP-GNN), to gather node's information at a distance of $L$, we need set the number of layers equal to $L$, hence the space complexity is $O(nL)$ and time complexity is $O(n \cdot L)$. As for pruned WL test(pruned MP-GNN), to gather node's information at a distance of $L$, we only need set the number of layers at most $[\log(L)] + 1$, hence the space complexity is $O(n \cdot \log(L))$ and time complexity is $O(n \cdot \log(L))$

$K$**-Path Framework**: For $K$-path WL test ($K$-Path GNN) to gather node's information at a distance of $L$, we need set the number of layers equal to $\frac{L}{K}$, at every layer, $K$-path WL aggregate from 1-Path neighbor to $K$-Path neighbor, therefore the space complexity and time complexity for per layer are $O(n \cdot K)$ $O(n \cdot K)$, the space complexity and time complexity for whole model are $O(n \cdot L)$ and $O(n \cdot L)$. As for pruned $K$-Path WL test (pruned $K$-Path GNN), the number of layers is set equal to $\frac{L}{K}$ too. Since $(n \gg L \gg K)$, at most layers, the architecture of pruned $K$-Path WL test is same as WL test, the space complexity and time complexity for whole model are $O(n \cdot \frac{L}{K})$ and $O(n \cdot \frac{L}{K})$.

$K$**-Hop Framework**:

Same as $K$-Path Framework, at every layer, $K$-Hop WL aggregate from 1-Path neighbor to $K$-Path neighbor, the space complexity and time complexity for whole model are $O(n \cdot L)$ and $O(n \cdot L)$. For pruned $K$-Hop WL test (pruned $K$-Hop GNN), the space complexity and time complexity for whole model are $O(n \cdot \frac{L}{K})$ and $O(n \cdot \frac{L}{K})$.

## G. Consistency of Expressiveness for Pruned WL Test

To verify the expressive power equivalence between the pruned k-path WL, the WL test and their original algorithms, while for K-Hop WL, WHETHER loses information about non-shortest paths is critical. We conducted experiments on both real-world and synthetic datasets. For graph tasks, we perform the original WL test on each graph to assign a classification label to every node. Subsequently, we execute the pruned WL test on the same graph. If the pruned WL test produces the same number of node classes as the original WL test($|\{C_{WL}^i\}| = |\{C_{PRWL}^i\}| = m$), and there exists a mapping $f : [m] \to [m]$ between the classes that $|\{v|\chi(v) \in C_{WL}^{f(i)}\}| = |\{v|\chi(v) \in C_{PRWL}^i\}|$, then we conclude that the pruned WL test is consistent with the original WL test on that graph. we execute both the original and pruned WL tests on each individual graph for node-level tasks. The WL Test algorithm applies the hashed and power iterated color refinement algorithm proposed by Kersting (Kersting et al., 2014), and others are derived from it. $\pi(\cdot)$ denotes the prime number function($\pi(i)$ the $i^{th}$ prime), in multiple aggregations WL test we assign different primes into different aggregation $(\pi((K - l) \cdot C^{(l-1)} + k \cdot \mathbf{1}))$. This avoids hash collision since if $k$ is different $(\pi((K - l) \cdot C^{(l-1)} + k \cdot \mathbf{1}))$ will be different. The algorithm is as follow:

### G.1. Real-World

As shown in Table 9, we conducted comparative experiments between WL and pruned WL on real-world datasets, Acc denotes the proportion that the pruned variant output coincides with original WL test. Time(s) denotes the time of cost, and Aver denotes the average number of classes that algorithms assign to node in graph.

---

**Algorithm 8** HCGCR: Hashed CGCR(WL Test Algorithm)

---

1: **Input:** Adjacency matrix $A$, number of iteration $L$.
2: **Output:** Nodes corresponding class $C^{(L)}$, number of class $m^{(L)}$
3: **Initialization:** $C^{(0)} \leftarrow \mathbf{1}$, $m^{(0)} \leftarrow 1$
4: **for** $l = 1$ to $L$ **do**
5: $\quad C^{(l)} \leftarrow \text{unique}(C^{(l-1)} + A\log(\pi(C^{(l-1)})))$
6: $\quad m^{(l)} \leftarrow \max(C^{(l)})$
7: **end for**

---

---

**Algorithm 9** Pseudocode for the Pruned WL Test Algorithm

---

1: **Input:** Adjacency matrix $A$, number of aggregations of each iteration $\mathbf{T}$, number of aggregation for each iteration $L$
2: **Output:** Nodes corresponding class $C^{(L)}$, number of class $m^{(L)}$
3: **Initialization:** $C^{(0)} \leftarrow \mathbf{1}$, $m^{(0)} \leftarrow 1$
4: **for** $l = 1$ to $L$ **do**
5: $\quad C_0^{(l-1)} \leftarrow \log(\pi(C^{(l-1)}))$
6: $\quad$ **for** $t = 1$ to $\mathbf{T}[\mathbf{l}]$ **do**
7: $\quad\quad C_t^{(l-1)} \leftarrow A \cdot C_{t-1}^{(l-1)}$
8: $\quad$ **end for**
9: $\quad C^{(l)} \leftarrow \text{unique}(C^{(l-1)} + C_{\mathbf{T}[\mathbf{l}]}^{(l-1)})$
10: $\quad m^{(l)} \leftarrow \max(C^{(l)})$
11: **end for**

---

---

**Algorithm 10** Pseudocode for the $K$-Hop(Path) WL Test Algorithm

---

1: **Input:** $K$, k-Hop(Path) Adjacency matrix $A_k(k \in [K])$, number of iteration $L$
2: **Output:** Nodes corresponding class $C^{(L)}$, number of class $m^{(L)}$
3: **Initialization:** $C^{(0)} \leftarrow \mathbf{1}$, $m^{(0)} \leftarrow 1$
4: **for** $l = 1$ to $L$ **do**
5: $\quad$ **for** $k = 1$ to $K$ **do**
6: $\quad\quad C_k^{(l-1)} \leftarrow A_K \cdot \log(\pi((K-1) \cdot C^{(l-1)} + k \cdot \mathbf{1}))$
7: $\quad$ **end for**
8: $\quad C^{(l)} \leftarrow \text{unique}(C^{(l-1)} + \sum_{k \in [K]} C_k^{(l-1)})$
9: $\quad m^{(l)} \leftarrow \max(C^{(l)})$
10: **end for**

---

---

**Algorithm 11** Pseudocode for the Pruned $K$-Hop(Path) WL Test Algorithm

---

1: **Input:** $K$, k-Hop(Path) Adjacency matrix $A_k(k \in [K])$, number of iteration $L$
2: **Output:** Nodes corresponding class $C^{(L)}$, number of class $m^{(L)}$
3: **Initialization:** $C^{(0)} \leftarrow \mathbf{1}$, $m^{(0)} \leftarrow 1$
4: **for** $l = 1$ to $L$ **do**
5: $\quad$ **if** $l \leq K - 1$ **then**
6:
7: $\quad\quad$ **for** $k = l$ to $K$ **do**
8: $\quad\quad\quad C_k^{(l-1)} \leftarrow A_k \cdot \log(\pi((K-l) \cdot C^{(l-1)} + k \cdot \mathbf{1}))$
9: $\quad\quad\quad C^{(l)} \leftarrow \text{unique}(C^{(l-1)} + \sum_{k \in [l,K]} C_k^{(l-1)})$
10: $\quad\quad$ **end for**
11: $\quad$ **else**
12: $\quad\quad C_K^{(l-1)} \leftarrow A_K \cdot \log(\pi(C^{(l-1)}))$
13: $\quad\quad C^{(l)} \leftarrow \text{unique}(C^{(l-1)} + C_K^{(l-1)})$
14: $\quad$ **end if**
15: $\quad m^{(l)} \leftarrow \max(C^{(l)})$
16: **end for**

---

*Table 9.* Pruned WL Test on real-world dataset.

| Model | MUTAG | | | PTC | | | NCI1 | | | PROTEINS | | |
|---|---|---|---|---|---|---|---|---|---|---|---|---|
| | Acc | Time | Aver | Acc | Time | Aver | Acc | Time | Aver | Acc | Time | Aver |
| $WL^{G.1}$ | 100% | 1.522 | 11.196 | 100% | 4.265 | 11.069 | 100% | 139.715 | 19.227 | 100% | 98.185 | 31.622 |
| $PRWL^{G.1}$ | | 0.557 | 11.196 | | 1.328 | 11.069 | | 31.207 | 19.227 | | 11.340 | 31.622 |
| $WL^{G.1}$ | 98.9% | 14.644 | 14.739 | 100% | 31.912 | 15.075 | 99.5% | 631.695 | 40.376 | 100% | 363.649 | 37.286 |
| $PRWL^{G.1}$ | | 3.454 | 14.771 | | 8.624 | 15.075 | | 145.192 | 40.319 | | 127.640 | 37.286 |
| $WL^{G.1}$ | 98.9% | 25.536 | 14.803 | 100% | 77.879 | 15.392 | 99.5% | 916.559 | 48.259 | 100% | 779.075 | 41.986 |
| $PRWL^{G.1}$ | | 7.134 | 14.795 | | 20.049 | 15.392 | | 191.903 | 48.452 | | 178.697 | 41.986 |
| $K-Hop^{G.1}$ | 100% | 2.175 | 16.161 | 100% | 6.093 | 13.394 | 100% | 199.593 | 44.995 | 100% | 166.914 | 41.015 |
| $PR2-Hop^{G.1}$ | | 1.903 | 16.161 | | 5.331 | 13.394 | | 174.644 | 44.995 | | 147.712 | 41.015 |
| $2-Hop^{G.1}$ | 100% | 10.976 | 17.077 | 100% | 30.845 | 16.150 | 100% | 295.236 | 51.155 | 100% | 290.919 | 43.174 |
| $PR2-Hop^{G.1}$ | | 10.192 | 17.077 | | 28.642 | 16.150 | | 274.148 | 51.155 | | 276.373 | 43.174 |
| $2-Path^{G.1}$ | 100% | 2.240 | 16.324 | 100% | 6.275 | 13.529 | 100% | 205.581 | 45.449 | 100% | 171.922 | 41.429 |
| $PR2-Path^{G.1}$ | | 1.960 | 16.324 | | 5.491 | 13.529 | | 179.883 | 45.449 | | 152.143 | 41.429 |
| $2-Path^{G.1}$ | 100% | 11.306 | 17.249 | 100% | 31.770 | 16.314 | 100% | 304.093 | 51.672 | 100% | 299.646 | 43.610 |
| $PR2-Path^{G.1}$ | | 10.498 | 17.249 | | 29.501 | 16.314 | | 282.372 | 51.672 | | 284.664 | 43.610 |

The parameter settings for the number of layers are configured as follows:(1)WL(MP-GNN):3 layers,7 layers and 10 layers.(2)Pruned WL:[1,2]layers,[1,2,4]layers,[1,2,3,4]layers.(3)K-Hop:K=2,3 layers,:K=2,5 layers.(4)Pruned K-Hop: K=2,3 layers,K=2,5 layers.(5)K-Path: K=2,3 layers,K=2,5 layers.(6)Pruned K-Path: K=2,3 layers,K=2,5 layers.

### G.2. Synthetic

As shown in Table 10, we conducted comparative experiments between WL and pruned WL on Synthetic datasets for node-level, Equi denotes the whether the pruned variant output coincides with original WL test. Time denotes the time of cost, and class denotes the number of classes that algorithms assign to node in graph. Every graph has 200 nodes, and RANDOM is a random graph with edge generated probability 0.01, based on the principle of preferential attachment (rich-get-richer effect) with edge generated probability 0.01, and 3-Random Regular is a regular graph where each node has 3 degrees.

The experimental results lead us to the following conclusions:(1)The pruned algorithm almost perfectly matches the original algorithm, the few mismatches occur because CUDA operations on high-precision data introduce computational errors, leading the algorithm to (with low probability) classify WL-equivalent nodes into different categories.(2)The complexity of pruned algorithm is stricly reduced, especially for pruned WL algorithm.(3)Under the same K value and number of layers, the K-Path method yields more class number than K-Hop. This aligns with the fact that K-Hop pruning loses information about non-shortest paths. (4)Under the condition of aggregating identical distances greater than K, the pruned K-Hop/Path WL demonstrates stronger expressive power than WL (classifying nodes into more categories) while requiring less computation time than WL.(5)The pruned 2-Hop achieves complete expressive equivalence with the original algorithm, demonstrating that pruned K-Hop can be considered approximately equal to standard K-Hop in terms of expressive power.

### G.3. Ablation on Complexity

Table 11 presents a comparison between the pruned and original models in terms of training time, where Acc denotes model accuracy and Time represents the duration required for one training iteration. Table 12 demonstrate the comparison between total training time and aggregation time, where the aggregation time accounts for a relatively small proportion of the total time. The pruned models achieve comparable accuracy to the original model, while most pruned variants demonstrate better training efficiency than the baseline.

*Table 10.* Synthetic Data

| Model | RANDOM | | | Barabási–Albert | | | 3-Random Regular | | |
|---|---|---|---|---|---|---|---|---|---|
| | Equi | Time $(s)$ | class | Equi | Time $(s)$ | class | Equi | Time $(s)$ | class |
| $WL^{G.1}$ | ✓ | 0.468 | 121 | ✓ | 0.515 | 130 | ✓ | 0.015 | 1 |
| $PRWL^{G.1}$ | | 0.026 | 121 | | 0.028 | 130 | | 0.014 | 1 |
| $WL^{G.1}$ | ✓ | 3.585 | 145 | ✓ | 3.934 | 155 | ✓ | 0.034 | 1 |
| $PRWL^{G.1}$ | | 0.721 | 145 | | 0.794 | 155 | | 0.030 | 1 |
| $WL^{G.1}$ | ✓ | 7.327 | 153 | ✓ | 8.009 | 163 | ✓ | 0.051 | 1 |
| $PRWL^{G.1}$ | | 1.878 | 153 | | 2.064 | 163 | | 0.046 | 1 |
| $K-Hop^{G.1}$ | ✓ | 0.585 | 141 | ✓ | 0.644 | 151 | ✓ | 0.380 | 12 |
| $PR2-Hop^{G.1}$ | | 0.568 | 141 | | 0.625 | 151 | | 0.369 | 12 |
| $2-Hop^{G.1}$ | ✓ | 2.868 | 153 | ✓ | 3.149 | 163 | ✓ | 0.837 | 23 |
| PR 2-Hop 5 | | 2.940 | 153 | | 3.228 | 163 | | 0.812 | 23 |
| $2-Path^{G.1}$ | ✓ | 0.591 | 141 | ✓ | 0.651 | 151 | ✓ | 0.384 | 15 |
| $PR2-Path^{G.1}$ | | 0.574 | 141 | | 0.631 | 151 | | 0.373 | 15 |
| $2-Path^{G.1}$ | ✓ | 2.894 | 153 | ✓ | 3.178 | 163 | ✓ | 0.846 | 26 |
| $PR2-Path^{G.1}$ | | 10.498 | 153 | | 3.257 | 163 | | 0.820 | 26 |

*Table 11.* Comparison on training efficiency

| Model | COLLAB | | NCI1 | | IMDB-B | | IMDB-M | | MUTAG | | PROTEINS | |
|---|---|---|---|---|---|---|---|---|---|---|---|---|
| | Time | Acc (%) | Time | Acc (%) | Time | Acc (%) | Time | Acc (%) | Time | Acc (%) | Time | Acc (%) |
| GIN(3) | 1.104 | $74.8 \pm 1.3$ | 0.480 | $71.9 \pm 0.5$ | 0.251 | $71.9 \pm 0.3$ | 0.304 | $49.9 \pm 0.0$ | 0.889 | $89.4 \pm 0.4$ | 0.268 | $73.7 \pm 0.7$ |
| PR GIN(1) | 1.060 | $73.9 \pm 0.0$ | 0.461 | $72.9 \pm 1.4$ | 0.209 | $69.9 \pm 2.0$ | 0.284 | $50.6 \pm 0.3$ | 0.886 | $88.5 \pm 0.0$ | 0.233 | $72.2 \pm 1.9$ |
| GIN(7) | 1.638 | $77.4 \pm 1.6$ | 0.748 | $71.5 \pm 1.4$ | 0.578 | $72.6 \pm 0.3$ | 0.534 | $51.1 \pm 0.3$ | 0.904 | $89.4 \pm 1.0$ | 0.527 | $76.3 \pm 0.2$ |
| PR GIN(124) | 1.284 | $76.4 \pm 0.7$ | 0.6961 | $75.4 \pm 0.2$ | 0.481 | $71.7 \pm 1.4$ | 0.464 | $52.0 \pm 0.5$ | 0.916 | $92.0 \pm 0.4$ | 0.425 | $74.1 \pm 1.0$ |
| GIN(10) | 2.142 | $74.7 \pm 0.6$ | 1.122 | $75.9 \pm 1.3$ | 0.948 | $72.1 \pm 2.8$ | 0.898 | $49.7 \pm 1.5$ | 0.874 | $87.7 \pm 0.2$ | 0.867 | $72.3 \pm 0.0$ |
| PR GIN(1234) | 1.689 | $75.6 \pm 0.3$ | 0.981 | $74.7 \pm 0.2$ | 0.710 | $71.5 \pm 0.5$ | 0.780 | $51.2 \pm 0.9$ | 0.929 | $90.7 \pm 2.1$ | 0.667 | $72.5 \pm 2.2$ |
| 2-Hop(3) | 1.357 | $76.8 \pm 0.8$ | 0.608 | $73.6 \pm 0.9$ | 0.410 | $71.0 \pm 0.7$ | 0.415 | $50.1 \pm 1.5$ | 0.910 | $91.0 \pm 0.0$ | 0.394 | $69.5 \pm 1.3$ |
| PR 2-Hop(3) | 1.180 | $75.1 \pm 1.1$ | 0.528 | $76.5 \pm 1.6$ | 0.357 | $71.5 \pm 0.5$ | 0.361 | $52.5 \pm 0.7$ | 0.929 | $91.3 \pm 1.5$ | 0.342 | $73.3 \pm 0.3$ |
| 2-Hop(5) | 1.927 | $74.2 \pm 0.5$ | 0.880 | $70.6 \pm 1.7$ | 0.680 | $68.8 \pm 0.8$ | 0.628 | $49.5 \pm 0.6$ | 0.894 | $88.3 \pm 1.0$ | 0.621 | $71.0 \pm 0.8$ |
| PR 2-Hop(5) | 1.606 | $74.5 \pm 0.4$ | 0.734 | $71.1 \pm 1.5$ | 0.566 | $69.7 \pm 0.2$ | 0.523 | $48.0 \pm 2.2$ | 0.871 | $88.7 \pm 1.5$ | 0.517 | $72.0 \pm 0.3$ |
| 2-Path(3) | 1.385 | $75.6 \pm 0.0$ | 0.620 | $73.0 \pm 0.8$ | 0.418 | $71.4 \pm 0.1$ | 0.423 | $49.5 \pm 1.8$ | 0.9045 | $88.7 \pm 1.7$ | 0.402 | $72.7 \pm 2.0$ |
| PR 2-Path(3) | 1.385 | $76.1 \pm 0.3$ | 0.632 | $75.5 \pm 0.4$ | 0.488 | $74.2 \pm 1.9$ | 0.451 | $51.6 \pm 0.4$ | 0.915 | $91.6 \pm 0.0$ | 0.446 | $76.9 \pm 0.7$ |
| 2-Path(5) | 2.111 | $76.0 \pm 0.2$ | 0.964 | $74.2 \pm 0.3$ | 0.745 | $70.0 \pm 0.6$ | 0.688 | $51.3 \pm 0.1$ | 0.890 | $89.4 \pm 0.4$ | 0.680 | $69.3 \pm 1.0$ |
| PR 2-Path(5) | 1.730 | $72.1 \pm 2.0$ | 0.790 | $70.5 \pm 2.3$ | 0.610 | $71.8 \pm 1.5$ | 0.564 | $50.2 \pm 0.2$ | 0.898 | $88.9 \pm 0.8$ | 0.557 | $71.9 \pm 2.4$ |

*Table 12.* Comparison on aggregation time

| Model | COLLAB | | NCI1 | | IMDB-B | | IMDB-M | | MUTAG | | PROTEINS | |
|---|---|---|---|---|---|---|---|---|---|---|---|---|
| | Time | Agg. Time | Time | Agg. Time | Time | Agg. Time | Time | Agg. Time | Time | Agg. Time | Time | Agg. Time |
| GIN(3) | 1.104 | 0.091 | 0.480 | 0.033 | 0.251 | 0.017 | 0.304 | 0.021 | 0.889 | 0.063 | 0.268 | 0.018 |
| PR GIN(12) | 1.060 | 0.087 | 0.461 | 0.032 | 0.209 | 0.014 | 0.284 | 0.020 | 0.886 | 0.062 | 0.233 | 0.016 |
| GIN(7) | 1.638 | 0.143 | 0.748 | 0.052 | 0.578 | 0.040 | 0.534 | 0.037 | 0.904 | 0.064 | 0.527 | 0.037 |
| PR GIN(124) | 1.284 | 0.133 | 0.6961 | 0.049 | 0.481 | 0.033 | 0.464 | 0.032 | 0.916 | 0.065 | 0.425 | 0.029 |
| GIN(10) | 2.142 | 0.215 | 1.122 | 0.079 | 0.948 | 0.067 | 0.898 | 0.063 | 0.874 | 0.062 | 0.867 | 0.061 |
| PR GIN(1234) | 1.689 | 0.188 | 0.981 | 0.069 | 0.710 | 0.050 | 0.780 | 0.055 | 0.929 | 0.065 | 0.667 | 0.047 |
| 2-Hop(3) | 1.357 | 0.116 | 0.608 | 0.042 | 0.410 | 0.028 | 0.415 | 0.029 | 0.910 | 0.064 | 0.394 | 0.027 |
| PR 2-Hop(3) | 1.180 | 0.1 | 0.528 | 0.037 | 0.357 | 0.025 | 0.361 | 0.025 | 0.929 | 0.065 | 0.342 | 0.024 |
| 2-Hop(5) | 1.927 | 0.168 | 0.880 | 0.062 | 0.680 | 0.048 | 0.628 | 0.044 | 0.894 | 0.063 | 0.621 | 0.043 |
| PR 2-Hop(5) | 1.606 | 0.14 | 0.734 | 0.051 | 0.566 | 0.040 | 0.523 | 0.036 | 0.871 | 0.061 | 0.517 | 0.036 |
| 2-Path(3) | 1.385 | 0.118 | 0.620 | 0.043 | 0.418 | 0.029 | 0.423 | 0.029 | 0.9045 | 0.064 | 0.402 | 0.028 |
| PR 2-Path(3) | 1.385 | 0.12 | 0.632 | 0.044 | 0.488 | 0.034 | 0.451 | 0.031 | 0.915 | 0.064 | 0.446 | 0.031 |
| 2-Path(5) | 2.111 | 0.185 | 0.964 | 0.068 | 0.745 | 0.052 | 0.688 | 0.048 | 0.890 | 0.063 | 0.680 | 0.048 |
| PR 2-Path(5) | 1.730 | 0.151 | 0.790 | 0.055 | 0.610 | 0.043 | 0.564 | 0.039 | 0.898 | 0.063 | 0.557 | 0.039 |

