# OpenReview forum: "Pruning for GNNs: Lower Complexity with Comparable Expressiveness"
_ICML.cc/2025/Conference — ICML 2025 poster_

### Official Review · Reviewer_h6cq · 2025-03-04

**Overall Recommendation:** 3

**Summary:**

In this paper, the author proposed pruned MPNNs, K-path GNNs, and K-hop GNNs to reduce the computation redundancy in the original method. The author proved the expressive equivalence between the original version and the pruned version. The proposed method is evaluated on various benchmarking datasets and shows comparable results to the original version.

## Update after rebuttal

I would like to thank the authors for the detailed response to my questions and concerns. The author provides both a textual explanation for the complexity analysis of their method and the experimental comparison result. This helps to solve my biggest concern about the proposed method. I would like to increase my original score.

**Claims And Evidence:**

1. In the title, the authors claim that the pruned GNNs have higher expressiveness. However, I think it’s kind of misleading, as the expressive power of pruned GNNs is bounded by the original version based on the proofs provided in the paper. It is only safe to say that given the limited number of layers/iteration, the pruned version is better than the original one.
2. In Table 5, the authors claim that the pruned version has better complexity than the original version. However, I am not convinced. First, I do not understand why the space complexity is associated with the distance $L$. Since for both MPNN, K-path GNNs, and K-hop GNNs or their pruned version, we only need to save embedding for each node in the graph, which results in $O(n)$. For time complexity, it’s also not correct, as we need to consider the density of the graph. Let’s use the average degree $d$ to characterize. For MPNNs, it should be $O(ndL)$. For PR MPNNs, it seems to me that when we need to aggregate higher order neighbors, the complexity of the aggregation will go up, which means that the complexity is different for different layers. The first layer is just $nd$. For the second layer, we will aggregate neighbors from the second hop, which results in $nd^2$. Overall, it should be $O(\sum_{log(L)}^{i}nd^{2^{i-1}})$
. Similarly, the analysis for K-path and K-hop GNNs is also not correct to me. Given my understanding of the complexity part, I don’t think the proposed pruned framework will result in huge computation improvement over the original version. Even worse, the pruned version requires aggregation of different neighbors for different layers, which introduces additional pre-processing time and is not flexible for adjusting the number of layers.
3. The author mentioned that fewer layers can reduce the nonlinearity of GNNs and reduce over-smoothing. However, it is a problem primarily for node tasks. But the proposed methods are mainly used in Graph-level tasks.

**Essential References Not Discussed:**

The topic is related, and the experiment protocol is almost identical to the existing paper [1], but it is not cited, discussed, and compared.

[1] Jiarui Feng, et al., How powerful are K-hop message passing graph neural networks, NeurIPS22.

**Experimental Designs Or Analyses:**

1. I believe a more comprehensive ablation study on the complexity is **necessary** for showing the potential of the proposed methods. Specifically, I would like to see the comparison of the original version and the pruned version on both pre-process, train, and inference time under different numbers of layers and datasets (different graph distribution). Meanwhile, the cost of space should also be evaluated to support Table 5.
2. I would suggest another ablation study between the number of layers and the performance using both the original version and the pruned version using both the simulation data and real-world data to see if the pruned version really has better performance, especially under a imited number of layers.

**Methods And Evaluation Criteria:**

1. As mentioned above, although the proposed method requires fewer layers to achieve certain expressive power, the complexity is varied and much higher than the pruned version for each layer. Therefore, it is hard to see that the proposed method is more efficient.
2. The pruning strategy seems totally different for different GNNs. Are there any common criteria for pruning so that we can apply a similar strategy to other GNNs?
3. Even if the proposed method does introduce efficiency due to the fewer number of layers. The application is limited. Specifically, the advantage is $log(L)$ over $L$. However, real-world graph tasks are usually molecules, whose $L$ are usually less than 10. Therefore, it seems not a big improvement to me.

**Other Comments Or Suggestions:**

See above.

**Other Strengths And Weaknesses:**

See above.

**Questions For Authors:**

See above.

**Relation To Broader Scientific Literature:**

If the method really has a computation advantage over the existing method, it can be used in the field of molecular or biomedical research.

**Theoretical Claims:**

The complexity analysis seems wrong to me. See above. Given the limited time, I cannot check the proofs of all theorems. All other conclusions seem correct to me.

---

> ### Author Rebuttal · Authors · 2025-03-27
>
> We sincerely thank all the reviewers for their feedback and constructive comments
>
> Claims And Evidence:
>
> (2)Space Complexity:During backpropagation in training, the node representations of intermediate layers (e.g., activation values) are required to compute gradients and update weights. If intermediate node representations are discarded, backpropagation cannot proceed. Therefore node representations from each layer need to be cached, which results in $O(n·L)$.
>
> Time Complexity:We speculate the reviewer might draw conclusion:$\sum^i_{log(L)}O(nd^{2^{i-1}})$ due to the divergence in computational paradigms. Suppose aggreation method as sum. In the reviewer’s view, the aggregation of $l$ times is computed as follow:(1)Compute the $l$-th power of the adjacency matrix $A^l$.(2)Calculate the product of $A^l$ and node representations $h$ as $A^l·h$. This computation method involves calculating the product between matrices L times, resulting in extremely high complexity (even if A is a sparse matrix).
>
> However in our approch (mentioned in Equation (15) on page 5), the aggregation is decomposed into $l$ times of 1-neighborhood aggregation. In other word, $A^l·h$ is computed as:
>
>   $for$ $i$ $in$ $[l]$ $do$:  $h_i=A·h_{i-1}$
>
> Our approch will only compute  the product between matrice and node representations vector. Thus, it achieves lower computational complexity compared to the approach above. Therefore the complexity of aggregation is $\sum^i_{log(L)}O(2^{i-1}nd)=O(ndL)$. We provided pruned architectures better visualized in the link.
>
> As for reviewer's concern for complexity, we decompose the computational time complexity into two components: (1) feature aggregation($O_{agg}(1)$ refers to one basic aggreation for a node) and (2) the MLP operations($O_{MLP}(1)$ refer one layer's MLP) and space complexity ($O_{spa}(1)$ refers to one node's represetations). Table 5 can be modified into the following table(we assume K<<L):
>
> |Training Complexity | MP-GNN | PR MP-GNN |K-HOP|PR K-HOP|K-PATH|PR K-PATH|
> |----------|----------|----------|----------|----------|----------|----------|
> | Time  | $O_{agg}(nL)+O_{MLP}(L)$ | $O_{agg}(nL)+O_{MLP}(log(L))$|$O_{agg}(nL)+O_{MLP}(L/K)$ | $O_{agg}(nL/K)+O_{MLP}(L/K)$ | $O_{agg}(nL)+O_{MLP}(L/K)$ | $O_{agg}(nL/K)+O_{MLP}(L/K)$ |
> | Space   | $O_{spa}(nL) $  | $O_{spa}(nlog(L)) $ | $O_{spa}(nL/K) $  | $O_{spa}(nL/K) $  | $O_{spa}(nL/K) $  | $O_{spa}(nL/K) $  |
>
>  On the other side, in our ablation experiment, the time consumed by aggregation is not significant.
>
> As for adjusting the number of layers, $a_l$ doesn's have to be $2^{l-1}$, we point out $2^{l-1}$ because it will reach its maximum expressiveness. Theorem4.1has shown as long as it is viewable, it will be as powerful as WL. Hence we can flexibly adjust $a_l$  as geometric sequence($a_l=l$) or other viewable sequence.
>
> (1):The Initial intention of the title derives from the comparasion of MP-GNN and PR K-PATH GNN instead of MP-GNN and PR MP-GNN: from(2), if K<<L, the complexity of PR K-PATH GNN is less than MP-GNN, while PR K-PATH is as powerful as K-PATH and K-PATH is stirctly more powerful than MP-GNN. We sincerely appreciate the reviewer's insightful comment regarding the potential ambiguity in the paper's title. In response to the reviewer's suggestion, we would like to modify the title to "PRUNING For GNNs: LOWER COMPLEXITY WITH COMPARABLE EXPRESSIVENESS" to better reflect the study's focus.
>
> (3)The pruned methods can also be used in Node-level tasks,  since the graph representation is a aggreation of node representations. In other word, given a Pruned MP-GNN $M'$ as powerful as arbitrary MP-GNN $M$, if for any two nodes they get same representation $M$ it will also get same representation in $M'$.
>
> Methods And Evaluation Criteria:
>
> (1)It has been mentioned above
>
> (2) The pruning strategy seems different because MP-GNN is one-aggreation while K-HOP\PATH is Multi-aggregation. Our pruned method  for MP-GNN is suitable for any one-aggreation GNN, but pruned method for Multi-aggregation is associations between different aggregated neighbor nodes (can not guarantee the expressive equivalence).
>
> (3)The reason why GNNs typically aggregate node information within a distance L≤10 is that larger L values lead to issues such as excessive parameter size (prone to overfitting on small datasets) and high computational/memory costs. However, our pruning method reduces the parameters to a logarithmic scale (log(L)),thereby potentially enabling GNNs to aggregate information from more distant nodes
>
>
> Experimental Designs Or Analyses:We have supplemented the ablation experiments on pruning efficiency and conducted experiments on the large-scale graph OGB-arXiv. The experimental conclusion can be found in our response to other reviewers. All the experimental materials(code), results, the better visualized pruned architectures and responses to other questions are provided in https://anonymous.4open.science/r/PrunedGNN-AC61/README.md

---

> > ### Comment · Reviewer_h6cq · 2025-04-02
> >
> > I would like to thank the authors for the detailed clarification and response. Most of my concerns have been addressed. I will increase my score accordingly. Please include all the additional results in the future version.

---

> > > ### Author Response · Authors · 2025-04-06
> > >
> > > Thank you sincerely for your generous feedback and for raising your evaluation of my work. I greatly appreciate the time and thought you invested in reviewing my manuscript and offering such valuable suggestions. Your support is truly meaningful, and I’ve found the revision process very rewarding thanks to your insights.

---

### Official Review · Reviewer_V1ZW · 2025-03-14

**Overall Recommendation:** 3

**Summary:**

The paper proposes a pruning framework for GNNs aimed at improving computational efficiency while maintaining or even enhancing expressive power. The authors introduce pruned versions of Message Passing GNNs (MP-GNNs), K-Path GNNs, and K-Hop GNNs by identifying and removing redundant structures. Theoretical analysis using MATLANG demonstrates that pruned versions retain the expressive power of their unpruned counterparts, and experiments on multiple datasets validate the efficiency gains. The main contributions include:

- A theoretical justification for pruning in GNNs without sacrificing expressive power.
- A pruned message passing framework that maintains equivalent expressiveness while reducing complexity.
- Empirical validation showing that pruned frameworks outperform or match unpruned models across benchmark datasets with lower computational cost.

**Claims And Evidence:**

The authors claim that:
- Pruning redundant structures does not reduce expressive power - This is well-supported by theoretical analysis using MATLANG and empirical validation on graph isomorphism tasks.
- Pruned models have lower computational complexity - The paper provides clear evidence through algorithmic complexity analysis and runtime measurements.
- Pruned K-Hop and K-Path GNNs distinguish more non-isomorphic graphs than MP-GNNs - While theoretically sound, this claim could benefit from more empirical results on diverse graph structures.
- Pruning improves training efficiency and scalability - The results support this claim, but more large-scale graph evaluations (e.g., OGB datasets) would strengthen it.

Overall, the claims are mostly well-supported, but additional empirical validation, particularly on large-scale datasets, would reinforce the conclusions.

**Essential References Not Discussed:**

The following studies could be cited, as they present different pruning techniques that can be compared against the proposed approach in the paper:

[1] Dupty et al., PF-GNN: Differentiable particle filtering based approximation of universal graph representations, ICLR 2022.

[2] Fey et al., GNNAutoScale: Scalable and Expressive Graph Neural Networks via Historical Embeddings, ICML 2021.

[3] Müller et al., GraphChef: Decision-Tree Recipes to Explain Graph Neural Networks, ICLR 2024.

**Experimental Designs Or Analyses:**

The experiments are well-designed but could be improved in a few areas:

- Expressiveness experiments: The graph isomorphism tests effectively demonstrate the theoretical claims.
- Graph property prediction: Results on the TU datasets and QM9 confirm that pruning does not degrade performance.
- Computational efficiency: The paper successfully demonstrates reduced parameter count and runtime improvements.

However, experiments on larger graphs (e.g., OGB datasets) and ablation studies on different pruning strategies would strengthen the empirical claims.

**Methods And Evaluation Criteria:**

The methodology is well-structured, with clear mathematical definitions and derivations. The pruning strategies are rigorously developed, and the expressiveness analysis is grounded in matrix algebra. The evaluation criteria include:

- Expressiveness tests: Distinguishing non-isomorphic graphs.
- Graph property prediction: Evaluating node and graph features.
- Real-world benchmarks: TU datasets, QM9, and ZINC. The chosen benchmarks and evaluation metrics are appropriate, though additional experiments on larger-scale datasets would provide further validation.

**Other Comments Or Suggestions:**

Please refer to the weaknesses mentioned above.

**Other Strengths And Weaknesses:**

**Strengths**
- The paper introduces a novel pruning strategy for GNNs, which reduces computational complexity while preserving expressive power. Unlike existing methods that enhance expressiveness by increasing depth or complexity, this approach optimizes efficiency without sacrificing performance.
- The paper is well-structured, with clear theoretical justifications, proofs, and empirical results supporting the claims.
- The extensive experimentation across synthetic and real-world datasets strengthens the findings, particularly for efficiency improvements.
- If widely adopted, the proposed pruning strategies could make expressive GNNs more computationally feasible for large-scale applications, such as molecular modeling.

**Weaknesses**
- While the theoretical analysis supports the expressiveness claims, the experimental validation relies mostly on synthetic datasets and indirect comparisons (e.g., isomorphism tests). Direct comparisons on real-world tasks requiring high expressiveness (e.g., molecular property prediction) would strengthen the argument. A more comprehensive ablation study showing how different pruning levels affect expressiveness would be useful.
- The efficiency claims are theoretically sound, but the experimental validation on large-scale graphs is somewhat lacking. How well do the pruned frameworks generalize to datasets with millions of nodes and edges?
- In particular, K-Hop pruning loses information about non-shortest paths. While the authors argue that this is not critical, further empirical evidence would help support this claim.

**Questions For Authors:**

1. Are there any known cases where pruned K-Hop fails to distinguish graphs that the original K-Hop framework can differentiate? Providing examples would help clarify the limitations.
2. The claim that pruning redundant structures enhances efficiency without compromising expressiveness is central to the paper. Could the authors conduct an ablation study comparing different pruning strategies to determine whether all identified redundant structures should be removed?
3. Have the authors tested how pruning affects GNNs trained on very large graphs (e.g., OGB datasets)? If so, do the training and inference times scale as expected?
4. The experimental results validate the effectiveness of pruned frameworks but do not directly measure expressiveness beyond isomorphism-based tasks. Could the authors evaluate the frameworks on tasks that require high expressiveness, such as molecular property prediction with long-range dependencies?
5. Could there be datasets where pruning negatively impacts expressiveness? If so, an analysis of when pruning is beneficial versus detrimental would strengthen the conclusions.

**Relation To Broader Scientific Literature:**

The paper is well-grounded in the existing literature on GNN expressiveness and efficiency:

- Expressiveness limits of MP-GNNs (Xu et al., 2019; Morris et al., 2019) - The pruning approach aligns with previous findings that message-passing GNNs are constrained by 1-WL limitations.
- Higher-order GNNs (Maron et al., 2019; Azizian & Lelarge, 2021) - The authors position their work as a computationally efficient alternative to higher-order approaches.
- Subgraph-based methods (Bevilacqua et al., 2022; Zhao et al., 2022) - The paper could better contrast pruning with subgraph-based improvements in expressiveness.

**Theoretical Claims:**

The paper provides several key theoretical results:

- Proof that pruned MP-GNNs retain 1-WL equivalence - The proof is logically structured and appears correct.
- Equivalence of pruned K-Path GNNs to standard K-Path GNNs - This is rigorously shown via MATLANG.
- Pruned K-Hop GNNs retain equivalence for distinguishing regular graphs - The proof is incomplete due to lost structural information.
- Computational complexity analysis - The theoretical derivations appear sound.

---

> ### Author Rebuttal · Authors · 2025-03-31
>
> We sincerely acknowledge reviewer V1ZW for the constructive criticism and insightful suggestions.
>
> Questions For Authors:
>
>  Q1: Unfortulately, we have to admit that find a pair of graphs that pruned K-Hop fails to distinguish graphs and the original K-Hop framework can differentiate involves a tremendous amount of algebraic analytic work. Since the proof of the theorem of the existance of a pair of graphs distinguishable by the (k+1)-dimensional Weisfeiler-Lehman ((k+1)-WL) test but not by the k-dimensional Weisfeiler-Lehman (k-WL) test spans 58 pages. However, even if it exists, we can draw the conclusion that the probability for inconsistent expressiveness between K-Hop and  pruned K-Hop goes to 0 as the size of graphs n goes to infinity. As shown in the respond to Q2, we also conducted experiments on this issue real-world and synthetic datasets, all of the results show that they have same expressiveness.
>
> Q2: We have added experiments to verify the consistency of the expressive power between the pruned architecture and the original algorithm on both real and synthetic datasets. The experimental results show that the expressive power of the pruned architecture is identical to that of the original algorithm. We also conducted ablation experiments for GNNs, result shows the pruned version achieves significant reductions in both parameter count and training duration, showing noticeable efficiency improvements.
>
> We have supplemented ablation experiments analyzing the impact of different pruning strategies on model efficiency and accuracy.
> We summarize the core conclusion:(1)Regarding to the expressiveness(Graph Isomorphism WL test), all identified redundant structures should be removed since it will enhance efficiency.(2)When it comes to the performances(accuracy), sometimes retaining certain redundant structures can slightly improve the model's accuracy. The reason is that if all identified redundant structures are removed, the subsequent layers will need to aggregate a large amount of representations. Since MLP are not strict hash functions, this can degrade the model's performance.
>
> As for pruning strategies for MP-GNNs, we point out that the sequence $a_l$ can be flexibly adjusted, as long as $a_l$ is viewable(The subset's sum of {$a_l$} is dense in $[S_l]$), such as geometric sequence($a_l=l$) or fibonacci sequence($a_l=a_{l-1}+a_{l-2}$), then it have same expressiveness as MP-GNN.
>
> Q3:We have tested how pruning affects GNNs trained on very large graphs on dataset obg-arvix. The result shows that  while maintaining comparable accuracy to the original model, the pruned version achieves significant reductions in both parameter count and training duration, showing noticeable efficiency improvements.
>
> Q4:As shown in Q2, we have conducted WL Test on both real and synthetic datasets. For the pruned WL algorithm, it is considered accurate only when its graph output results are entirely consistent with the original algorithm. This requires high expressiveness. Meanwhile, experimental results demonstrate a significant improvement in the efficiency of the pruned algorithm.
>
> Q5:During the experiments, the performance for 4 layers [1,2,3,4] Pruned GIN often performs worse than  3 layers [1,2,3] Pruned GIN, We find out the reason is， when the layer gets deeper，a substantial number of neighbor representations will be repeatedly  aggregated. This significantly impedes the model's ability to extract useful information. Consequently, we refined the model to eliminate redundant representation aggregation, thereby enhancing Pruned GIN's performance.
>
> Weaknesses:In particular, K-Hop pruning loses information about non-shortest paths. While the authors argue that this is not critical, further empirical evidence would help support this claim.
>
> Our original intention was to point out that, compared to the K-Path framework, K-Hop loses information about non-shortest paths. As a result, we cannot prove that the pruned K-Hop is equivalent in expressive power to the original model. However, this does not have a significant impact. Firstly, we demonstrate the equivalence between the pruned K-Hop and the original model for regular graphs and strongly regular graphs—while the K-Hop model was specifically designed to address the inability of MP-GNNs to distinguish regular graphs. Secondly, in our ablation experiments, the pruned K-Hop exhibits equivalent expressive power to the original model for both real-world and Synthetic experiments.
>
> Compared to K-Path,  K-Hop indeed is less powerful than K-Path, since K-Path asigned more number of classes during the WL expressiveness experiment. All the experimental materials (code), results and responses to References in provided in https://anonymous.4open.science/r/PrunedGNN-AC61/README.md
>
> Thanks again, for the reviewer's detailed analysis and questions, which allowed me to further refine the model in the ogbn-arxiv experiments.

---

> > ### Comment · Reviewer_V1ZW · 2025-04-02
> >
> > Thank you for your considerate answers to my questions. I truly appreciate the time and effort you took to address my concerns. I will be keeping my original score.

---

> > > ### Author Response · Authors · 2025-04-06
> > >
> > > I’m very grateful for your thoughtful comments. Your feedback significantly contributed to refining the manuscript, and I truly value your support throughout the revision process.

---

### Official Review · Reviewer_hERx · 2025-03-14

**Overall Recommendation:** 2

**Summary:**

This paper proposes pruned versions of Message Passing GNNs, K-Hop GNNs, and K-Path GNNs by eliminating redundant structures. The authors claim that these pruned frameworks maintain or even improve expressive power while reducing computational complexity. Theoretical analysis based on matrix language is used to demonstrate equivalence in expressiveness between the pruned and original frameworks. Additionally, experimental results on benchmark datasets show that pruned GNNs achieve comparable or better performance with improved efficiency.

**Claims And Evidence:**

The paper claims that pruning redundant structures in MP-GNNs, K-Hop GNNs, and K-Path GNNs maintains expressiveness while reducing complexity. While some theoretical arguments support this claim, the practical significance is not well justified.

**Essential References Not Discussed:**

No significant gaps found.

**Experimental Designs Or Analyses:**

The experimental setup evaluates pruned GNNs on several benchmark datasets. However, the improvements in efficiency are marginal, and the comparisons do not convincingly demonstrate practical benefits over existing optimized GNN models. I do not think the TUdatasets are enough to evaluate the method. The authors should consider OGB datasets for evaluations.

**Methods And Evaluation Criteria:**

The methodology is theoretically sound in using matrix language tools to analyze expressiveness. However, the evaluation mainly focuses on standard benchmark datasets without strong ablation studies or comparisons with more sophisticated recent baselines.

**Other Comments Or Suggestions:**

Consider adding experiments on larger, real-world datasets to better illustrate the efficiency gains.

**Other Strengths And Weaknesses:**

The theoretical analysis provides an interesting perspective on expressiveness equivalence. But the empirical evaluations are limited.

**Questions For Authors:**

Have you tested the approach on large-scale graphs where efficiency improvements might be more noticeable?

**Relation To Broader Scientific Literature:**

It improves the expressiveness of GNNs (broader literature) while reducing complexity.

**Theoretical Claims:**

The theoretical analysis claims that pruned GNNs maintain the expressive power of the original architectures.

---

> ### Author Rebuttal · Authors · 2025-03-31
>
> We thank reviewer hERx careful evaluation and meaningful suggestions.
>
> Q1:The methodology is theoretically sound in using matrix language tools to analyze expressiveness. However, the evaluation mainly focuses on standard benchmark datasets without strong ablation studies or comparisons with more sophisticated recent baselines.
>
> To verify the expressive power equivalence between the pruned WL test and their original algorithms, We have added experiments to verify the consistency of the expressive power between the pruned architecture and the original algorithm on both real and synthetic datasets. The experimental results show that the expressive power of the pruned architecture is almost identical to that of the original algorithm.
>
> Additionally, we have evaluated the efficiency improvements of the pruned architecture compared to the original architecture in the WL algorithm and as a GNN model. The results show that under the WL algorithm, the pruned architecture maintains the same expressive power as the original algorithm while significantly reducing computation time. As for GNN, We compared the pruned model in terms of accuracy and training time, and the results show that the pruned models achieve comparable accuracy to the original model, while most pruned variants demonstrate better training efficiency than the baseline. Some of the results are shown below.
>
> | Model         | COLLAB Time   |COLLAB Acc (%) | NCI1 Time   | NCI1 Acc (%) | IMDB-B Time   | IMDB-B Acc (%) | IMDB-M Time   | IMDB-M Acc (%) | MUTAG Time   | MUTAG Acc (%) |  PROTEINS Time   | PROTEINS Acc (%) |
> |---------------|--------|---------|--------|---------|--------|---------|--------|---------|--------|---------|--------|---------|
> | GIN(3)        | 1.104  | 74.8 ± 1.3 | 0.480  | 71.9 ± 0.5 | 0.251  | 71.9 ± 0.3 | 0.304  | 49.9 ± 0.0 | 0.889  | 89.4 ± 0.4 | 0.268  | 73.7 ± 0.7 |
> | PR GIN(1)    | 1.060  | 73.9 ± 0.0 | 0.461  | 72.9 ± 1.4 | 0.209  | 69.9 ± 2.0 | 0.284  | 50.6 ± 0.3 | 0.886  | 88.5 ± 0.0 | 0.233  | 72.2 ± 1.9 |
> | GIN(7)        | 1.638  | 77.4 ± 1.6 | 0.748  | 71.5 ± 1.4 | 0.578  | 72.6 ± 0.3 | 0.534  | 51.1 ± 0.3 | 0.904  | 89.4 ± 1.0 | 0.527  | 76.3 ± 0.2 |
> | PR GIN(124)  | 1.284  | 76.4 ± 0.7 | 0.6961 | 75.4 ± 0.2 | 0.481  | 71.7 ± 1.4 | 0.464  | 52.0 ± 0.5 | 0.916  | 92.0 ± 0.4 | 0.425  | 74.1 ± 1.0 |
> | GIN(10)       | 2.142  | 74.7 ± 0.6 | 1.122  | 75.9 ± 1.3 | 0.948  | 72.1 ± 2.8 | 0.898  | 49.7 ± 1.5 | 0.874  | 87.7 ± 0.2 | 0.867  | 72.3 ± 0.0 |
> | PR GIN(1234) | 1.689  | 75.6 ± 0.3 | 0.981  | 74.7 ± 0.2 | 0.710  | 71.5 ± 0.5 | 0.780  | 51.2 ± 0.9 | 0.929  | 90.7 ± 2.1 | 0.667  | 72.5 ± 2.2 |
> | 2-Hop(3)    | 1.357  | 76.8 ± 0.8 | 0.608  | 73.6 ± 0.9 | 0.410  | 71.0 ± 0.7 | 0.415  | 50.1 ± 1.5 | 0.910  | 91.0 ± 0.0 | 0.394  | 69.5 ± 1.3 |
> | PR 2-Hop(3) | 1.180  | 75.1 ± 1.1 | 0.528  | 76.5 ± 1.6 | 0.357  | 71.5 ± 0.5 | 0.361  | 52.5 ± 0.7 | 0.929  | 91.3 ± 1.5 | 0.342  | 73.3 ± 0.3 |
> | 2-Hop(5)    | 1.927  | 74.2 ± 0.5 | 0.880  | 70.6 ± 1.7 | 0.680  | 68.8 ± 0.8 | 0.628  | 49.5 ± 0.6 | 0.894  | 88.3 ± 1.0 | 0.621  | 71.0 ± 0.8 |
> | PR 2-Hop(5) | 1.606  | 74.5 ± 0.4 | 0.734  | 71.1 ± 1.5 | 0.566  | 69.7 ± 0.2 | 0.523  | 48.0 ± 2.2 | 0.871  | 88.7 ± 1.5 | 0.517  | 72.0 ± 0.3 |
> | 2-Path(3)    | 1.385  | 75.6 ± 0.0 | 0.620  | 73.0 ± 0.8 | 0.418  | 71.4 ± 0.1 | 0.423  | 49.5 ± 1.8 | 0.9045 | 88.7 ± 1.7 | 0.402  | 72.7 ± 2.0 |
> | PR 2-Path(3) | 1.385  | 76.1 ± 0.3 | 0.632  | 75.5 ± 0.4 | 0.488  | 74.2 ± 1.9 | 0.451  | 51.6 ± 0.4 | 0.915  | 91.6 ± 0.0 | 0.446  | 76.9 ± 0.7 |
>
> Q2:The experimental setup evaluates pruned GNNs on several benchmark datasets. However, the improvements in efficiency are marginal, and the comparisons do not convincingly demonstrate practical benefits over existing optimized GNN models. I do not think the TUdatasets are enough to evaluate the method. The authors should consider OGB datasets for evaluations.
>
> After we made appropriate improvements to the models to make them suitable for large-scale graphs, we conducted large-scale graph experiments on dataset ogb-arvix for GIN, Pruned GIN, and Pruned multiple aggregation GIN, the results show that  while maintaining comparable accuracy to the original model, the pruned version achieves significant reductions in both parameter count and training duration, showing noticeable efficiency improvements. Some of the results are shown below.
>
> |Model|Test Accuracy|Val Accuracy|Parameter |Total time|
> |----------|----------|----------|----------|----------|
> |GIN|0.7012 $\pm$ 0.0114|0.7132$\pm$0.0023|2.29M |2.23h|
> |Pruned GIN |0.6905 $\pm$ 0.0241|0.7231$\pm$0.0041|0.98M |1.52h||
> |Pruned 2-Mul GIN|0.7121 $\pm$ 0.0092|0.7341$\pm$0.0066|1.14M |1.73h|
>
> And all the experimental materials(code) and results in provided in https://anonymous.4open.science/r/PrunedGNN-AC61/README.md

---

### Decision · Program_Chairs · 2025-05-01

**Decision:**

Accept (poster)

**Comment:**

The paper proposes pruning techniques for various graph neural network architectures, including Message Passing (MP), K-Path, and K-Hop GNNs. The core contribution is a theoretically grounded framework for pruning redundant structures within these models, which reduces complexity while preserving the expressive power relative to traditional architectures. The authors provide rigorous theoretical justifications using matrix language (MATLANG) and substantiate their claims through experimental validations. The authors should include all the rebuttal efforts into the revision, especially more experiments on large-scale datasets to demonstrate the practical effectiveness and efficiency of the pruned versions. Other than that, it is known that theoretical expressiveness does not always turn into practical distinguishability of non-isomorphic graphs. Thus, I strongly encourage the authors to test their pruned GNNs on the BREC dataset (https://arxiv.org/pdf/2304.07702) to test whether the pruning compromises practical expressiveness.